



A two-decades (1988-2009) record of diatom fluxes in the Mauritanian coastal upwelling:

Impact of low-frequency forcing and a two-step shift in the species composition

Oscar E. Romero[1, 2,*], Simon Ramondenc[1,2] and Gerhard Fischer[1,3]

[1]MARUM – Center for Marine Environmental Sciences, University of Bremen, Leobener Str. 8, 28359 Bremen, Germany

[2]Alfred Wegener Institute, Helmholtz Centre for Polar and Marine Research, 27568 Bremerhaven, Germany

[3]University of Bremen, Geosciences Department, Klagenfurter Str., 28359 Bremen, Germany

*E-mail corresponding author: oromero@marum.de; oromero@uni-bremen.de

***Key words:*** coastal upwelling, decadal, diatoms, Eastern Boundary Upwelling Ecosystems, fluxes, Mauritania, multiyear, northwest Africa, sediment traps, time-series

**Abstract**

Eastern Boundary Upwelling Ecosystems (EBUEs) are among the most productive marine regions in the world's oceans. Understanding the degree of interannual to decadal variability in the Mauritania upwelling system is key for the prediction of future changes of primary productivity and carbon sequestration in the Canary Current EBUE as well as in similar environments. A multiannual sediment trap experiment was conducted at the mooring site CBmeso (='Cape Blanc mesotrophic', ca. 20°N, ca. 20°40'W) in the high-productive Mauritanian coastal waters. Here, we present results on fluxes and the species-specific composition of the diatom assemblage for the time interval between March 1988 and May 2009. The temporal dynamics of diatom populations allows to propose three main diatom productivity/flux intervals: (*i*) early 1988 - late 1996; (*ii*) 1997 - 1999, and (*iii*) early 2002 - mid 2009. The impact of the Atlantic Multidecadal Oscillation appears to be an important forcing of the long-term dynamics of diatom population. The impact of cold (1988-1996) and warm AMO phases (2001-2009) is reflected by the outstanding shifts in species-specific composition. This AMO-impacted, long-term trend is interrupted by the occurrence of the strong 1997 ENSO. The extraordinary shift in the relative abundance of benthic diatoms in May 2002 suggests the strengthening of offshore advective transport within the uppermost layer of filament waters, and in the subsurface and in deeper and bottom-near layers. It is hypothesized that the dominance of benthic diatoms was the response of the diatom community to the intensification of the slope and shelf poleward undercurrents. This dominance followed the intensification of the warm phase of AMO and the associated changes of the Atlantic Meridional Overturning Circulation. Transported valves (siliceous remains) from shallow coastal waters into the deeper bathypelagial should be considered for the calculation and model





experiments of bathy- and pelagial nutrients budgets (especially Si), the burial of diatoms and the

        paleoenvironmental signal preserved in downcore sediments. Our 1988-2009 data set contributes to

        the distinction between climate-forced and intrinsic variability of populations of diatoms and will be

especially helpful for establishing the scientific basis for forecasting and modelling future states of

        this ecosystem and its decadal changes.

**1 Introduction**

        As part of the latitudinally extended Eastern Boundary Upwelling Ecosystem (EBUE) of the Canary

        Current (CC) in the subtropical northeastern Atlantic, the Mauritanian upwelling is characterized by

intense offshore Ekman transport and strong mesoscale heterogeneity. This physical setting

        facilitates the vigorous exchange between the neritic and pelagic realms off Mauritania (Chavez and

        Messié, 2009; Cropper et al., 2014; Freón et al., 2009). The nutrient trapping efficiency of upwelling

cells (Arístegui et al., 2009), the input of wind-carried dust particles from the Sahara and the Sahel

        (Fischer et al., 2016; Fischer et al., 2019; Friese et al., 2016; Romero et al., 1999, 2003), and/or the

        wide shelf (Hagen, 2001; Cropper et al., 2014) additionally impact the intensity of the primary

production in surface waters and the subsequent export of microorganism remains into the

        bathypelagial off Mauritania. This set of conditions varies strongly on different temporal patterns

        (from seasonal through decadal; Mittlelstaedt, 1983, 1991; Hagen, 2001; Nykjær and Van Camp,

1994; Barton et al., 2013; Varela et al., 2015). Whether the strong interannual and decadal variability

        of physical conditions off Mauritania is related to low frequency, global-scale climatic variations or to

        an intrinsic level of basin-wide atmospheric and/or oceanic variability is still a matter of debate

(Cropper et al., 2014, Fischer et al., 2016, 2019; Varela et al., 2015).

        EBUEs may prove more resilient to on-going climate change than other ocean ecosystems

        because of their ability to function under extremely variable conditions (Barton et al., 2013; Varela et

al., 2015). On the other hand, it is predicted that current global warming will impact atmospheric

        pressure gradients and hence the strength of coastal winds that cause upwelling (Bakun, 1990; Bakun

        et al., 2013). Although productivity variations in EBUEs are sensitive to the amplitude and timing of

upwelling-favorable winds (Varela et al., 2015), the impact of on-going ocean warming on the

        dynamics of upwelling-favorable winds is still contentious (Bakun, 1990; Bakun et al., 2013; Varela et

        al., 2015). Long-term trends in variations of upwelling intensity and related productivity changes

seem highly dependent on the length of the data series, the selected study area, the season

        evaluated, and the methods applied (Varela et al., 2015). The description of multiyear to interdecadal

        trends of upwelling intensity in the CC-EBUE has been mostly based on variations of velocity and

direction of winds and calculated upwelling intensities. Cropper et al. (2014) found a non-significant

        increase in upwelling-favorable winds along the CC-EBUE between 11° and 35°N. Using the same





database as Cropper et al. (2014), Narayan et al. (2010) and Patti et al. (2011) analyzed the annual

wind stress over four decades and found significant increasing trends across 24°–32°N. Contradictory

results were also obtained using Ekman transport data. Gómez-Gesteira et al. (2008) detected a

significant decreasing trend in upwelling intensity across 20°–32°N for all seasons between 1967 and

2006, while Pardo et al. (2011) found a general weakening of upwelling intensity between 10 and

24°N for the time interval 1970–2009. Barton et al. (2013) found no statistically significant change of

the annual mean wind intensity off Northwest Africa over the second half of the 20th century.

A different approach for the characterization of multiyear to interdecadal trends in EBUEs is

assessing fluxes of particulates and microorganisms as captured by continuous sediment trap

experiments. This study builds on earlier investigations of multiyear variability of the diatom flux

captured with sediment traps deployed at the mesopelagial mooring site CBmeso (=`Cape Blanc

mesotrophic´, formerly known as CB, Fig. 1; Fischer et al., 1996). Several earlier studies addressed

either the variations of total diatom fluxes between 03/1988 and 11/1991 (Lange et al., 1998;

Romero et al., 1999a, 2002; Romero and Armand, 2010) or the land-derived signal of siliceous

remains (Romero et al., 1999b, 2003). After a gap of 2,5 years (12/91 through 05/94), the CBmeso

trap experiment re-started in June 1994 (Table 1). Here, we extend the diatom record collected from

early June 1994 until middle June 2009. The main goal of this study is the description of the multiyear

dynamics of the total diatom flux (TDF) and the shifts in the species-specific composition of the

assemblage at site CBmeso during almost 20 years (1988-2009). To our knowledge, our study

presents the longest sediment trap-based time-series on the temporal dynamics of diatom fluxes in a

low-latitude EBUE. We discuss the new results in view of the high-frequency atmospheric and

hydrographic dynamics along the CC-EBUE, and low-frequency climate variability in the North

Atlantic, and compare our new dataset at site CBmeso with previous diatom (Romero and Fischer,

2017; Lange et al., 1998; Romero et al., 1999a,b; 2002, 2020), as well as bulk flux results off

Mauritania (Fischer et al., 2016, 2019; Helmke et al., 2005). We also discuss our new results with

recent results from the near-by, coastal site CBeu (='Cape Blanc eutrophic') (Romero and Fischer,

2017; Romero et al., 2020).

**2  Material and Methods**

**2.1  Mooring location,  sampling intervals and sample treatment**

A total of 20 moorings were deployed off Mauritania (ca. 21°N, 20°W, Fig. 1) between March 1988

and May 2009. Details on sampling intervals and trap depths are given in Table 1. Major gaps in the

diatom record are between: (*i*) December 1991 and June 1994 (no traps deployed), (*ii*) October 1994

and November 1995 (malfunctioning of the trap CBmeso6 upper); (*iii*) October 1997 and June 1998

(malfunctioning of the trap CBmeso8 upper), and November 1999 and March 2001 (malfunctioning

of traps CBmeso10 and 11 lower) (Table 1).



We used deep-moored (>1000 m depth), large-aperture, time-series sediment traps of the Kiel and Honjo types with 20 cups and 0.5 m² openings, equipped with a honeycomb baffle (Kremling et al., 1996). As the traps were moored in deep waters (mostly below 1,200m, Table 1), uncertainties with the trapping efficiency due to strong currents (*e.g.* undersampling) and/or due to the migration and activity of zooplankton migrators ('swimmer problem') are assumed to be minimal (Buesseler et al., 2007). Prior to the deployments, the sampling cups were poisoned with $HgCl_2$ (1 ml of conc. $HgCl_2$ per 100ml of filtered seawater) and pure NaCl was used to increase the density in the sampling cups to 40‰. Upon recovery, samples were stored at 4°C on board and wet-splitted in the home laboratory (Marum, University of Bremen) using a rotating McLane wet splitter system. Larger swimmers –such as crustaceans– were handpicked at the home lab by using forceps and were removed by filtering carefully through a 1 mm sieve. All flux data here refer to the size fraction <1 mm. In almost all samples, the fraction of particles >1 mm was negligible, only in a few samples larger pteropods were found (Fischer et al., 2016).

We compare out data with those previously published at the mooring location CBeu (ca. 20°45'N, 18°45'W), also deployed off Mauritania (Romero and Fischer, 2017; Romero et al., 2020). It locates ca. 80 nautical miles (∼150 km) offshore over the continental slope in ca. 2,750 m water depth. For site CBeu, only the upper trap fluxes are shown (Romero and Fischer, 2017; Fischer et al., 2019; Romero et al., 2020).

### 2.2 Assessment of diatom fluxes and species identification

1/64 and 1/125 splits of the original samples were used. Samples were rinsed with distilled water and prepared for diatom studies following standard methods (Schrader and Gersonde, 1978). For this study, a total of 282 sediment trap samples was processed. Each sample was chemically treated with potassium permanganate, hydrogen peroxide (33%) and concentrated hydrochloric acid (32%) following previously used methodology (Romero and Fischer, 2017; Romero et al., 1999a, b, 2002, 2009a, b; 2020). Qualitative and quantitative analyses of the diatom community were carried out on permanent slides of acid cleaned material (*Mountex®* mounting medium) at x1000 magnifications by using a *Zeiss®* Axioscop with phase-contrast illumination (Marum, University of Bremen). Depending on valve abundances in each sample, several traverses across each slide were examined. The counting procedure and definition of counting units for valves follows Schrader and Gersonde (1978). Total amount of counted valves per slide ranged between ca. 400 and 1,000. Two cover slips per sample were scanned in this way. Counts of valves in replicate slides indicate that the analytical error of valve concentration estimates is ≤10 %.

The resulting counts yielded abundance of individual diatom taxa as well as daily fluxes of valves per $m^{-2}$ $d^{-1}$ (DF), calculated according to Sancetta and Calvert (1988), as follows:

$$DF = \frac{[N] \times [A/a] \times [V] \times [Split]}{}$$



[days] x [D]

where, [N] number of valves in an known area [a], as a fraction of the total area of a petri dish [A]
and the dilution volume [V] in ml. This value is multiplied by the sample split [Split], representing the
fraction of total material in the trap, and then divided by the number of [days] of sample deployment
and the sediment trap collection area [D].

**2.3 Statistical analysis**

Correspondence Analysis (CA) was used to explore diatoms community's changes. CA is an
ordination technique that enables describing the community structure from multivariate contingency
tables with frequency-like data (*i.e.* abundances derived from counting with integers and zeros) that
are dimensionally homogeneous (Legendre and Legendre, 2012). Based on the CA samples' scores, a
hierarchical clustering analysis was performed to classify the samples date according to the diatom
composition similarities. Euclidean distance was used to compute the distance matrix from which a
hierarchical dendrogram was generated using Ward's aggregation link (Legendre and Legendre,
2012). This approach has been computed by using the *vegan* package included in the R software. In
addition, Kruskall Wallis tests, coupled with multiple comparison tests (pairwise Wilcoxon rank sum
test) have been performed on climatic indexes and TDF according to sample groups highlighted by
the clustering analysis with the aim of identifying relationships between environmental forcing
indices and diatom communities.

**3 Physical setting of the study area**
**3.1 Oceanography, winds, and upwelling dynamics**

The CC-EBUE is in the eastern part of the North Atlantic subtropical gyre (Fig. 1; Arístegui et al.,
2009; Chavez and Messié, 2009; Cropper et al., 2014). Both the temporal occurrence and the
intensity of the upwelling along northwestern Africa depend on the shelf width, the seafloor
topography, wind direction and strength (Mittelstaedt, 1983; Hagen, 2001), the Ekman-mediated
transport and strong mesoscale heterogeneity (Chavez and Messié, 2009; Cropper et al., 2014; Fréon
et al., 2009). The Mauritanian shelf is wider than the shelf northward and southward along the CC-
EBUE and gently slopes from the coastline into water depths below 200 m (Hagen, 2001). The shelf
break zone with its steep continental slope extends over approximately 100 km from the coastline
(Hagen, 2001). As a consequence of the coastal topography, and the shelf and slope bathymetry, the
ocean currents and the wind system, surface waters off Mauritania are characterized by almost
permanent upwelling with varying intensity year-through (Lathuilière et al., 2008; Cropper et al.,
2014). Site CBmeso locates at the westward end of this permanent upwelling zone.

The surface hydrography off Mauritania is strongly influenced by two surface currents: the
southwestward-flowing CC and the poleward-flowing coastal countercurrent or Mauritania Current





(MC) (Fig. 1). The surficial CC detaches from the northern African continental slope between 25° and 21°N and supplies Si-poor waters to the North Equatorial Current. CC waters are relatively cool because it entrains upwelled water from the coast as it moves southward (Mittelstaedt, 1991). The

Si-rich MC gradually flows northward along the coast up to about 20°N (Mittelstaedt, 1991), and brings warmer surface waters from the equatorial realm into waters overlying site CBmeso. Towards late autumn, the MC is gradually replaced by a southward flow associated with upwelling water due

to the increasing influence of trade winds south of 20°N (Zenk et al., 1991), and becomes a narrow strip of less than 100 km width in winter (Mittelstaedt, 1983). The MC advances onto the shelf in summer and is enhanced by the relatively strong Equatorial Countercurrent and the southerly trade

winds (Mittelstaedt, 1983).

North of Cape Blanc (ca. 21°N; Fig. 1), the intense northeasterly winds cause coastal upwelling to move further offshore and the upper slope fills with upwelled waters. South of Cape Blanc, northerly

winds dominate year-through, but surface waters remain stratified and the coastal Poleward Undercurrent (PUC) occurs as a subsurface current (Pelegrí et al., 2017). South of Cape Timiris (ca. 19°30′N), the PUC intensifies during summer-fall and remains at the subsurface during winter–spring

(Pelegrí et al., 2017). The encountering of the northward flowing MC-PUC system with the southward flowing currents builds the Cape Verde Frontal Zone (Zenk et al., 1991; Fig. 1) and the large offshore water export is visible as the giant Mauritanian chlorophyll filament (Gabric, 1993; Pelegrí et al.,

2006, 2017).

The chlorophyll filament extends offshore up to 400 km (*e.g.*, Arístegui et al., 2009; Cropper et al., 2014; Van Camp et al., 1991), carrying a mixture of North and South Atlantic Central Water (NACW

and SACW, respectively) through an intense jet-like flow offshore (Meunier et al., 2012). Intense offshore transport acts an important mechanism for the export of cool, nutrient-rich shelf and upper slope waters. It has been estimated that this giant filament export about 50% of the coastal new

production offshore toward the open ocean during intervals of most intense upwelling (Gabric et al., 1993; Helmke et al., 2005; Lange et al., 1998; Van Camp et al., 1991). This transport impacts even more distant regions in the deep ocean, since sinking particles are strongly advected by lateral

transport in subsurface and deeper waters (Fischer and Karakaş, 2009; Fischer et al., 2009; Karakaş et al., 2006).

The SACW occurs in layers between 100 and 400 m depth the Banc d'Arguin and off Mauritania.

The hydrographic properties of upwelled waters over the shelf suggest that they ascend from depths between 100 and 200 m south off the Banc d'Arguin (Mittelstaedt, 1983). North of it, the SACW merges gradually into deeper layers (200-400 m) below the CC (Mittelstaedt, 1983). The biological

response is drastically accelerated in upwelled waters when the SACW of the upper part of the undercurrent feeds the onshore transport of intermediate layers to form mixed-water types on the



shelf (Zenk et al., 1991).

**3.2 Large-scale, low-frequency climate and oceanographic modes potentially affecting the Mauritanian upwelling area**

**3.2.1 Atlantic Multidecadal Oscillation (AMO)**: is the average of sea surface temperatures (SST) of
213 the North Atlantic Ocean (from 0° to 60°N, 80°W to 0°), detrended to isolate the natural variability
(Endfield et al., 2001). It is an on-going series of multidecadal cyclicity, with cool and warm phases
that may last 20-40 years at a time with a difference of about 15°C between extremes. These
216 changes are natural and have been occurring for at least the last 1,000 years. Since AMO is linked to
SST variations, it also plays a significant role in the decadal forcing of productivity changes (O'Reilly et
al., 2016). Fischer et al. (2016) state that the correlation of sea-level pressure with area-averaged (0–
219 70°N, 60–10°W) SST fluctuations over periods longer than 10 years highlights a center of action in the
tropical Atlantic with sea-level pressure reductions (weaker northeasterly winds) along with higher
Atlantic basin-wide sea-level pressure during a positive AMO phase. This shows the importance of
222 longer-term, Atlantic basin-scale SST variations for alongshore winds and upwelling trends at site
CBmeso.

Despite the indirect role for the atmosphere, the physical connection between the Atlantic
Meridional Overturning Circulation (AMOC) and the AMO is typically described in terms of oceanic
processes alone: since the AMOC transports heat northward over the entire Atlantic, an increase in
NADW formation should increase the strength of the AMOC, thus increasing oceanic meridional heat
transport convergence in the North Atlantic, resulting in a basin-scale warming of SSTs (Knight et al.,
2005). Atlantic Meridional Overturning Circulation (AMOC) variability itself is often attributed to
changes in North Atlantic Deep Water (NADW) formation due to anomalous Arctic freshwater fluxes
(Jungclaus et al., 2005) and/or atmospheric modes such as the North Atlantic Oscillation (*e.g.*,
Buckley and Marshall, 2016). In contrast, Clement et al. (2015) found that the pattern of AMO
variability can be reproduced in a model that does not include ocean circulation changes, but only
234 the effects of changes in air temperature and winds.

**3.2.2 El Niño/Southern Oscillation (ENSO) and La Niña**: is an irregularly periodic variation in winds
and SST over the tropical eastern Pacific Ocean, affecting the climate of much of the tropics and
237 subtropics of other ocean basins. The warming phase is known as El Niño and the cooling phase as La
Niña. The Southern Oscillation is the accompanying atmospheric component, coupled with the SST
variations. ENSO-related teleconnections in the CC-EBUE upwelling system have been described by
240 several authors (Behrenfeld et al., 2001; Pradhan et al, 2006; Zeeberg et al., 2008) and can be
illustrated by the negative correlation of sea level pressure with eastern tropical Pacific SST.

**3.2.3 North Atlantic Oscillation (NAO)**: characterizes the difference of atmospheric sea level
pressure between the Icelandic Low and the Azores High (Hurrell, 1995). These fluctuations control





the strength and direction of westerly winds and location of storm tracks across the North Atlantic. It varies over time with no periodicity. A positive phase of the NAO is associated with anomalous high pressure in the Azores high region and stronger northeasterly winds along the NW African coast. Especially during the months of November through April, the NAO is responsible for much of the variability of weather in the North Atlantic region, affecting wind speed and wind direction changes, changes in temperature and moisture distribution and the intensity, number and track of storms. Correlations during winter show that NAO and ENSO may have opposite effects on the CC-EBUE/northeastern Atlantic realm, for instance on wind fields, and consequently on upwelling with potential implications for deep ocean mass fluxes (Fischer et al., 2016).

## 4 Results

### 4.1 Total diatom fluxes

Marine diatoms are the main contributors to the siliceous fraction in samples collected with the CBmeso traps between March 1988 and June 2009. Silicoflagellates and radiolarians are secondary components of the siliceous fraction (not shown here), with minor contribution of land-derived freshwater diatoms and phytoliths. In term of number of individuals, the TDF was always one order to four orders of magnitude higher than that of the other siliceous organisms.

The TDF ranged $2.7 \times 10^3$-$3.3 \times 10^6$ valves m$^{-2}$ d$^{-1}$ (average = $4.0 \times 10^5$ valves ±$4.4 \times 10^5$), and shows strong interannual variability (Fig. 2). Highest fluxes (>$1.0 * 10^6$ valves m$^{-2}$ d$^{-1}$) occurred in 1988, late 1989, early 2002, 2003, late 2004, early 2005, early 2007 and early 2008. The lowest total diatom flux was recorded between 1997 and 1999 (range = $2.3*10^3$–$5.1*10^4$).

Maxima of TDF are defined here as those values that are higher than the TDF average $\pm 1$ standard deviation (STD) for the entire study period. Spring and summer show the higher amount of above-the-average TDF. Although the same number of maxima are recorded in fall as in summer and spring (n=17; spring, n=16), the absolute values of fall TDF maxima were predominantly lower than those of spring and summer. Winter has the lowest amount of TDF maxima (n=12).

### 4.2 Temporal variations of marine diatom populations

A total of 203 diatom species were identified in CBmeso samples between March 1988 and June 2009. To better understand the temporal variations of the diverse community, we follow the same grouping approach as already applied in the nearby trap site CBeu (Romero and Fischer, 2017; Romero et al., 2020). Out of 203 taxa, 109 species (whose average relative contribution is ≥0.50% for the entire studied interval) were distributed in four groups, according to the main ecological and/or habitat conditions they represent: (1) benthic, (2) coastal upwelling, (3) coastal planktonic, and (4) open-ocean diatoms. Taxa assigned to each group are listed in Table 2. Below the main species of each group and their main ecological significance are shortly described.



(1)   The benthic group is dominated by *Delphineis surirella*. As part of the epipsammic community*, D. surirella* is a benthic marine species that commonly thrives in the shallow euphotic zone of sandy shores, shelf and uppermost slope waters along temperate to cool seas, forming either

short or long chains of small valves (length=5-15 $\mu$m) (Andrews, 1981).

    (2)   The coastal upwelling group is composed by several species of *Chaetoceros* resting spores (RS) and the vegetative cells of *Thalassionema nitzschioides* var. *nitzschioides*. Both taxa are common

components of the coastal and hemipelagial upwelling assemblages in EBUEs (Romero and Armand, 2010; Abrantes et al., 2002; Nave et al., 2001; Romero et al., 2002). Vegetative cells of numerous *Chaetoceros* species (mainly those assigned to the section *Hyalochaete*, Rines and Hargraves, 1988)

rapidly respond to the weakening of upwelling intensity and nutrient depletion by forming endogenous resting spores, hence their high numbers in trap samples is interpreted to represent the strongest upwelling intensity (Romero and Armand, 2010; Abrantes et al., 2002; Nave et al., 2001;

Romero et al., 2002).

    (3)   Coastal planktonic species mostly thrive in neritic to hemipelagic, oligo-to-mesotrophic waters with moderate levels of dissolved silica (DSi). These species become more abundant during

intervals of decreased mixing, when upwelling weakens (Romero and Armand, 2010; Romero and Fischer, 2017; Crosta et al., 2012; Romero et al., 2009a; Romero et al., 2009b; Romero et al., 2020). The well-silicified species *Actinocyclus curvatulus*, *Cyclotella litoralis, Coscinodiscus radiatus* and

*Shionodiscus oestrupii* var. *venrickae* are the main contributors at the CBmeso site.

    (4)   Open-ocean taxa thrive in pelagic, oligotrophic, and warm to temperate waters with low siliceous productivity due to low DSi availability and weak mixing in surface waters (Romero and

Fischer, 2017; Nave et al., 2001; Romero et al., 2005; Crosta et al., 2012). This highly diverse group is dominated by several species of *Azpeitia*, together with *Fragilariopsis doliolus*, *Nitzschia bicapitata*, *Nitzschia interruptestriata*, *Roperia tesselata* and *Planktoniella sol*.

The multivariate analyses performed on the relative abundance of diatom populations (Fig. 3) confirms the strong interannual variability with significant shifts within the diatom community between 1988 and 2009. The first CA component covers 65.47% of the total variance and opposes

the samples dominated by benthic and coastal planktonic diatoms (Fig. 3a). The second CA axis explains 19.16% of the total variance and discriminate coastal upwelling and open ocean diatom. The clustering analysis allows the samples to be statistically grouped, and the complete time series was

segmented according to the four diatoms communities' affiliation (Fig. 3b). These clusters show clear changes of diatom populations' contribution throughout the time series (Fig. 3c) with the dominance of open-ocean and costal upwelling populations between 1988 and 1996. Open-ocean diatoms

dominated from 1997 to 2001, while benthic taxa were main contributors from 2002 to 2009.



A major shift in the relative contribution of the diatom groups is seen from May 2002 onward. This shift occurred in two steps (Fig. 2b). The percentage of benthic diatoms strongly increased

between middle May/early June 2002 (raise from 12.5 to 68.6%; trap CBmeso13, samples 2 and 3). Benthic diatom contribution decreased below 40% in early 2004. A second increase occurred in winter 2006, with values being mostly above 50% almost throughout until the end of the trap

experiment in June 2009 (Fig. 2b). The dominance of benthic diatoms at CBmeso also prevails after 2009 (Oscar E. Romero, unpublished data). The marked increase of variability of the benthic relative contribution is clearly evidenced by the highest variability among all diatom groups (1STD of each

group for the entire study is: (1) benthic = ±23.68%, (2) coastal upwelling = ±10.46%, (3) coastal planktonic = ±11.88% and (4) open-ocean = ±16.00%).

The impact of the environmental variables on diatom communities was investigated by simple

comparison using the samples clustering and the forcing values associated (Fig 4). AMO, the Shannon diversity index and TDF show significant differences between groups (Kruskall-Wallis test; *p-value<0.05*) whereas no statistical differences have been observed for the NAO, ENSO and Pacific

Decadal Oscillation indices. Only benthic diatoms (group 4) shown higher AMO values compared to the three other groups (pairwise Wilcoxon rank sum test; *p-value<0.05*). In addition, a Shannon diversity index gradient has been mainly observed with low, intermediate and high values in samples

dominate by benthic (*i.e.* group 4), coastal planktonic (*i.e.* group 3) and coastal upwelling/open-ocean (*i.e.* group 2/1) populations, respectively (pairwise Wilcoxon rank sum test; *p-value<0.05*). The statistical analysis also highlights that, during the dominance of the coastal upwelling populations,

the TDF was higher compared to TDF when other diatom group/s dominated the community. The correlogram performed between CA axes and the low-frequency climate indices also confirms these trends (not shown here). A significant positive and negative correlation has been found between the

first CA axis samples scores' with respectively AMO and Shannon diversity index (Fig. 3). Given that the first CA is positively driven by the benthic group, this confirms that the outstanding dominance of the benthic diatom *D. surirella* decreased the diversity, although it also seems to be promoted by

AMO strengthening. In the same way, the second CA axis samples scores are positively correlated with TDF, which confirms that coastal upwelling diatoms seems to promote the TDF.

**5 Discussion**

The long-term diatom record at site CBmeso offers the possibility of discussing population dynamics in the context of the high-frequency atmospheric and hydrographic dynamics along the CC-

EBUE, and low-frequency climate variability such in the North Atlantic. In 5.1, we discuss the impact of climate forcing on the long-term trends of the diatom community and the TDF, and the two-step shift in the species-specific composition of diatom populations. In the second subsection, we



compare the CBmeso data with those previously published at the hemipelagic site CBeu (Romero and

Fischer, 2017; Romero et al., 2020), and discuss (*i*) the effect of the giant chlorophyll filament and (*ii*)

the impact of lateral advection from the coastal area off Mauritania upon the hemi- and pelagial

along the NE Atlantic Ocean.

Based on outstanding shifts in the species-specific composition of the diatom assemblage

occurred throughout the study interval (Fig. 2b). We propose three main intervals in the multiyear

evolution of populations and discuss them in view of major environmental forcings: (*i*) early 1988 -

late 1996 (gradually decreasing trend of coastal upwelling diatoms); (*ii*) 1997–1999 (highest

contribution of open-ocean diatoms); and (*iii*) 2002–middle 2009 (major shift in the species-specific

composition: extraordinary increase and dominance of benthic diatoms).

**5.1  The impact of low frequency forcing on the variability of diatom populations off Mauritania**

**5.1.1  AMO and the two-step increase of benthic diatoms' contribution**

Based on the long-term trends of our data and their statistical analysis (Figs. 2-5), we suggest that

the proposed intervals were the response of the diatom populations to the impact of low frequency

forcing. As described above in 4.2, the benthic diatom community appears positively correlated with

AMO (Fig. 5). Among the low frequency forcings affecting the subtropical North Atlantic (see above

3.2), the AMO plays a key role in determining decadal variations of SST and meridional circulation

(*e.g.*, Knight et al., 2005; Wang and Zhang, 2013; McCarthy et al., 2015). It is widely accepted that

AMO is largely induced by AMOC variations and the associated fluctuations of heat transport

(Medhaug and Furevik, 2011; Wang and Zhang, 2013; Knight et al., 2005; McCarthy et al., 2015;

details in 3.2.1). Using observational data and model experiments, Wang and Zhang (2013) conclude

that the cooling of the subtropical North Atlantic (where the CBmeso is deployed) towards the end of

the warm AMO phase is largely due to the meridional advection by the anomalous northward

current. The anomalous cooling appears below 100 m and extend down to ca. 1,500 m water depth,

with a maximum cooling around 200 m between 8° and 20°N. During the cold phase of the AMO, the

anomalous southward meridional current is responsible for the subsurface ocean warming (Wang

and Zhang, 2013).

An additional effect of the AMO impact is the significant long-term weakening (strengthening) of

the gyre during warm (cold) phase of AMO. This weakening contributes to the anomalous northward

current in the subsurface (ca. 100-200 m), while its strengthening causes the anomalous southward

current. The decrease of upwelling diatoms' contribution between 1988 and 1996 (Fig. 2b) matches

the transition from a predominantly cool into a warm AMO phase during the late 1990s (Fig. 5; Wang

and Zhang, 2013, and references therein). The simultaneous increase in the contribution of open-

ocean diatoms is additional evidence for decreased diatom productivity (Fig. 2) and the predominant



occurrence of oligo-mesotrophic waters bathing the CBmeso site towards the earliest 1990s, with the stronger input of the silica-depleted NACW (see 3.1).

An extraordinary feature of the multiyear dynamics of diatom populations at the CBmeso site is the sharp shift in the species contribution (Fig. 2b). Linked to the intensification of the warm AMO phase into the latest 1990s, the anomalous northward MC strengthened the advection into open

waters (Wang and Zhang, 2013). These physical setting might have led to a stronger input of the DSi-rich SACW. The MC mixes into the giant Mauritanian chlorophyll filament and, thus, positively impacts diatom productivity by supplying DSi also during summer and fall, far earlier than the

typically wind-induced upwelling off Mauritania (mainly occurring in late winter-early spring, Mittelstaedt, 1983; Cropper et al., 2014).

The species shift into larger contribution of benthic diatoms follows a two-step increase pattern:

the first abrupt increase is observed in late May/early June 2002. The second increase occurs in winter 2006, with values mostly above 50% almost until the end of the trap experiment (June 2009, Fig. 4). The dominance of benthic taxa also prevails after afterward throughout until recently

recovered traps at site CBmeso (Oscar E. Romero, unpublished data). As already observed in previous studies at the more neritic site CBeu (Romero and Fischer, 2017; Romero et al., 2020; see further discussion in 5.2), the diatom *D. surirella* also dominates the benthic community at site CBmeso.

Since this small diatom (length=5-15 $\mu$m) mostly occurs attached to sand grains in shallow marine habitats within the euphotic zone and is occasional component of the thycoplanktonic community (Andrews, 1981), they originally thrive in shallow waters (above 50m depth) overlying the wide

Mauritanian upper shelf, are suspended and transported downslope until reaching the traps at the bathypelagial CBmeso site.

The intensification of the transport of AMOC intermediate waters during the warm phase of the

AMO (Wang and Zhang, 2013) might have also contributed to the strengthening of lateral transport from subsurface shelf waters upon the hemi- and bathypelagial off Mauritania. Earlier studies at the CBmeso time-series (Fischer et al., 2009, 2016) and observation-based model experiments conducted

along the Mauritanian upwelling (Helmke et al., 2005; Karakaş et al., 2006; Nowald et al., 2015) discussed already the role of intermediate and deep nepheloid layers in the lateral transport of particles and microorganisms remains upon the deeper bathypelagial. Based on the vigorous mixing

in the uppermost water column due to the confluence of northward and southward water masses and strong, predominantly westward winds off Mauritania (Fig. 1; see 3.1), the offshore transport from shallow into deeper waters is most intense between 20.5°N and 23.5°N along the northwestern

African margin. Erosional processes in the very dynamic coastal realm significantly contribute to the downward transport of particulates and microorganism remains (Meunier et al., 2012), and are responsible for sporadic particle clouds advected up to several hundreds of kilometers offshore



within intermediate and bottom-near nepheloid layers (Fischer and Karakaş, 2009; Fischer et al.,
2009, Nowald et al., 2015). This nepheloid layer-mediated transport additionally benefits from the
bathymetry of the Mauritanian shelf and slope (Nowald et al., 2015). The subsurface layer (100 to

300 m water depth), in turn strongly affected by the AMOC intensification due to AMO impact (Wang
and Zhang, 2013), might be the place of mixing processes of laterally-advected materials from the
shelf (where benthic diatoms predominantly thrive) by the activity of the giant chlorophyll filament,

with relatively fresh material derived from the open ocean surface (as represented by the other
three diatom groups). As the nepheloid layer-mediated transport contribute more intensively to the
deposition of diatom remains upon the lower slope and beyond than the direct vertical settling from

euphotic layer does after 2001, the area of final burial of diatom valves is effectively displaced from
their production area in surface waters overlying the CBmeso site into their area of final deposition in
deep-sea sediments below 4,000 m water depth.

**5.1.2  The occurrence of the strong 1997 ENSO and the response of the diatom community off
Mauritania**

        The long-term trends determined by the cold and warm phases of AMO was altered in the second

half of the 1990s by the impact of the strong 1997 ENSO (McPhaden, 1999). Although caution is
advised in the interpretation of the record due to a few gaps between 1996 and 1999 (Table 1), we
postulate that both low coastal upwelling diatom values (≤4 %)  and TDF between February 1997

and November 1999 (Fig. 2b) are the response of the diatom community to the impact of ENSO upon
the low-latitude NW Atlantic. The dominance of taxa predominantly related to waters of low-to-
moderate productivity (1997: highest contribution of open-ocean diatoms and lowest of coastal

upwelling diatoms; 1998-99: highest contribution of coastal planktonic and open-ocean diatoms,
typical of oligo-mesotrophic waters) evidences considerable changes in the physical setting of the
Mauritanian upwelling. Since the interval 1996-1999 records the lowest TDF for the entire study (Fig.

2a), we argue that ENSO negatively impacted on diatom productivity off Mauritania and most of the
total organic carbon captured with CBmeso traps (Fischer et al., 2016), was instead delivered either
by coccolithophorids or bacterioplankton.

A positive ENSO goes along with the weakening of E-NE winds off Mauritania (Pradhan et al.,
2006; Fischer et al., 2016). Weakened E-NE trades lead to the deepening of the thermocline below
the depth of the source of upwelled water, this hindering the mixing of the water column and

causing upwelling intensity off Mauritania to decrease until early 1998 (Pradhan et al., 2006).
Additionally, the size of the Mauritanian chlorophyll filament decreased between winter 1997 and
spring 1998, while became unusually large from autumn 1998 to spring 1999 (Fischer et al., 2009).

Aperiodic, pronounced decreases in the diatom flux in other ocean basins have been previously
associated with limiting nutrient levels due to ENSO-derived perturbations. The diatom production in



hemipelagial waters off northern Chile decreased extraordinarily during the strong 1997 ENSO

compared to earlier years (Romero et al., 2001). Similar negative impact assigned to ENSO

teleconnections have been observed in the California Current (Lange et al., 2000), in the Cariaco

Basin (Romero et al., 2009b) and in the pelagial Subarctic Pacific Ocean (Takahashi, 1987).

Complementary support of this ENSO-mediated impact on surface water productivity off

Mauritania is provided by variations of bulk biogenic fluxes at CBmeso. The almost 2.5 times higher

organic carbon flux during 1998-99 than in 1997 (Helmke et al., 2005) led to propose that, after

weakening due to impact of ENSO on the physical setting, upwelling intensified immediately

afterward during La Niña (Fischer et al., 2016). Similarly, the seasonal cycle of surface Chl-*a*

distribution in waters above the CBmeso site reveals a noticeable event (~250% increase) in

Mauritanian coastal waters (Pradhan et al., 2006).

**5.2  Comparison of diatom fluxes and populations' dynamics within the giant Mauritanian**

**chlorophyll filament (CBmeso *vs* CBeu)**

In this subsection, we compare the diatom flux and the assemblage composition at site CBmeso

with previous results from the nearby trap site CBeu gained between 2003 and 2009 (Romero and

Fischer, 2017; Romero et al., 2020). The CBeu site locates ca. 80 nautical miles (~150 km) offshore at

the continental slope below the giant Mauritanian chlorophyll filament, and hence between the

coastline and the outer CBmeso site (Fig. 1). These two trap locations are under different nutrient

availability and upwelling intensity between eutrophic (CBeu) and mesotrophic conditions (CBmeso)

(Romero and Fischer, 2017; Fischer et al., 2016, 2019).

The less favorable conditions for diatom productivity in waters overlying site CBmeso is evidenced

by lower TDF than at site CBeu. On the seasonal pattern, TDF at site CBmeso are always two orders

of magnitude lower than values gained at site CBeu (Fig. 6a). This also happens during fall, when the

highest average seasonal flux is recorded at CBmeso ($5.6*10^5$ valves m$^{-2}$ d$^{-1}$ *vs* $3.3*10^6$ valves m$^{-2}$ d$^{-1}$).

We advocate that these flux differences reflect (*i*) the offshore weakening of the transport via the

chlorophyll filament, (*ii*) the seaward decreasing concentration of nutrients within the filament

(Lathuilière et al., 2008; Meunier et al., 2012), (*iii*) the more intense upwelling in waters overlying the

Mauritanian slope (Mittelstaedt, 1983, 1991; Cropper et al., 2014), and (*iv*) the offshore weakening

of the lateral transport (Karakaş et al., 2006; Nowald et al., 2015). According to satellite imagery (Van

Camp et al., 1991; Gabric et al., 1993; Fischer et al., 2016), the CBmeso mooring locates only

occasionally beneath the giant chlorophyll filament and hence below nutrient-rich waters. In general,

the larger DSi availability (approximately 10 *vs*. 5 µM) and the higher Si:N ratios of the source waters

(SACW *vs*. NACW = 0.6 vs. 0.3; Arístegui et al., 2009) in coastal waters bathing site CBeu are reflected

in ca. threefold higher BSi fluxes at the coastal CBeu -whose fluxes are additionally affected by



stronger ballasting due to higher lithogenic input from northwestern Africa- compared to the offshore CBmeso (Fischer et al., 2019).

    Complementary support to the scenario of lower (higher) productivity levels at CBmeso (CBeu) is
provided by the species-specific composition of the assemblage: relative contribution of groups related with more oligo-mesotrophic waters is higher at CBmeso than at CBeu (coastal planktonic and open-ocean, Fig. 6d, e), while the opposite is true for diatoms typical of eutrophic waters (Fig.
6c). Despite the difference in the relative contribution, the species-specific composition of diatom groups is remarkably similar at both sites. All the main group taxa at site CBmeso (Table 2, see also 4.2) are also found in CBeu samples (see Table 2 in Romero and Fischer, 2017). This match makes
sense as both trap sites are linked via lateral advection through near-surface, intermediate and deeper nepheloid layers. In their earlier study, Romero and Fischer (2017) observed that the shift in the species composition at site CBeu toward a benthic-dominated assemblage occurred in early
winter 2006. Since benthic diatoms in the deeper CBmeso traps are transported via nepheloid layers from shallow coastal waters (see 5.1.1), the high percentage of benthic species at the CBmeso site (Fig. 6b) also evidences the impact of particulates derived from the Mauritanian inner shallow shelf
(Romero and Fischer, 2017; Fischer et al., 2009, 2016, 2019; Romero et al., 2020). The simultaneous occurrence of the second increase of benthic diatoms at CBmeso and the increase at the neritic site CBeu (Fig. 5) is a striking feature of the population shift over a large part of the Mauritanian
upwelling system. The transport of particulates and microorganism remains from their source in shallow coastal waters into the hemipelagic realm probably occurs within weeks (Karakaş et al., 2006, 2009). Phytoplankton thriving in Mauritanian surface waters can be transported as far as 400
507     km offshore from coastal waters (Gabric et al., 1993; Helmke et al., 2005; Barton et al., 2013). The MC might have helped in detaching benthic diatoms from their substrata (Romero and Fischer, 2017) and in transporting them northwestward into the hemipelagial/bathypelagial realm (where CBmeso
traps are deployed). These observations offer additional evidence of the impact of AMO via the strengthening of the meridional advection, the major nutrient input via the MC and the nepheloid layer-mediated transport into the deeper Mauritanian bathypelagial.

## 6 Conclusions

This multiyear diatom study offers an overall picture of the long-term evolution of diatom-based
productivity and fluxes and the response of diatom populations to the interaction of high- and low-frequency hydrographic and atmospheric forcing in the mid-latitude NE Atlantic Ocean. A unique, persistent trend in the long-term evolution of the TDF, either decreasing or increasing, is not
recognized in our ca. 20-year record.



The statistical analysis supports the proposed scenario of AMO as an important driver of diatom populations' dynamics off Mauritania. The correspondence of cold (1988-1996) and warm AMO phases (2001-2009) is reflected by the shift in species-specific composition. This overall trend is interrupted by the occurrence of the strong 1997 ENSO. Changes in the physical setting following 1997 ENSO (weakening of E-NE trade winds, thermocline deepening, weakened water column mixing) negatively impacted diatom fluxes off Mauritania.

Our CBmeso trap results allow corroborating that the abrupt shift in the assemblage composition occurred earlier off Mauritania (starting May 2002) than previously demonstrated (Romero and Fischer, 2017; Romero et al., 2020) and followed two steps. The two-step increase of benthic diatoms at the CBmeso site suggests that the intensification of the slope and shelf poleward undercurrents followed the intensification of the warm phase of AMO and the associated AMOC changes.

Diatom remains sink not only vertically off Mauritania, but they are also laterally advected from the shelf to the deeper bathypelagial via the nepheloid layer-mediated transport. Transported valves (siliceous remains) from shallow coastal waters into the deeper bathypelagial should be considered for the calculation and model experiments of bathy- and pelagial nutrients budgets (especially Si), the burial of diatoms and the paleoenvironmental signal preserved in downcore sediments.

Understanding the degree of interannual to decadal variability in the Mauritania upwelling system is key for the prediction of future changes of primary productivity along the NW African margin as well in other, economically important EBUEs. Our 1988-2009 data set might be instrumental in distinguishing between climate-forced and intrinsic variability of populations of primary producers (*e.g.*, diatoms) and are especially important for establishing the scientific basis for forecasting and modeling future states of this ecosystem and its decadal changes.

**Code and Data Availability**

Data are available at https://doi.pangaea.de/10.1594/PANGEAE.921237

**Author Contributions**

OER and GF devised the study. OER collected the data and wrote the manuscript. SR performed the statistical analysis. All authors contributed to interpretation and discussion of results.

**Competing Interests**





The authors declare that they have no conflict of interest.

**Acknowledgements**

We are greatly indebted to the masters and crews of the RVs Meteor, MS Merian, Polarstern and
Poseidon for several research expeditions along off Mauritania. Much appreciated is the help of the
RV Poseidon headquarters at Geomar (Klas Lackschewitz, Kiel, Germany) during the planning phases
of these research expeditions and the support by the German, Moroccan and Mauritanian
governments. The Institut Mauretanien de Recherches Océanographiques et des Pêches at
Nouadhibou (Mauritania) is acknowledged for its support and help in getting the necessary
permissions to perform our multiyear trap experiments in Mauritanian coastal waters. Götz Ruhland,
Nico Nowald and Marco Klann (MARUM, Bremen) were responsible for the mooring deployments
and home lab work. The long-term funding by the German Research Foundation (DFG) through SFB
261, the Research Center Ocean Margins (RCOM) and the current MARUM Excellence Cluster "The
Ocean in the Earth System" (University of Bremen, Bremen, Germany) made this study possible.





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





## Captions

TABLES

Table 1: Data deployment at site CBmeso (=Cape Blanc mesotrophic): trap name, coordinates

(latitude and longitude), ocean bottom depth, trap depth, sampling interval and sample amount.

Table 1                                                                                      Romero et al.

| Trap name | LAT | LONG | Bottom depth | Trap depth | Sampling | | nr. of samples | Previously published by |
|---|---|---|---|---|---|---|---|---|
| | N | W | m | m | start | end | | |
| **CBmeso1 lower** | 20°45.3' | 19°44.5' | 3646 | 2195 | 03/22/88 | 03/08/89 | 13 | Lange et al. (1998), Romero et al. (2002) |
| **CBmeso2 lower** | 21°08.7' | 20°41.2' | 4092 | 3502 | 03/15/89 | 03/24/90 | 22 | Romero et al. (2002) |
| **CBmeso3 lower** | 21°08.3' | 20°40.3' | 4094 | 3557 | 04/29/90 | 04/08/91 | 17 | Romero et al. (2002) |
| **CBmeso4 lower** | 21°08.7' | 20°41.2' | 4108 | 3562 | 03/03/91 | 11/19/91 | 13 | Romero et al. (2002) |
| **CBmeso5 lower** | 21°08.6' | 20°40.9' | 4119 | 3587 | 06/06/94 | 08/27/94 | 19 | |
| *CBmeso6 upper\** | 21°15.0' | 20°41.8' | 4137 | 771 | 09/02/94 | 10/16/94 | 2 | |
| **CBmeso7 lower** | 21°15.4' | 20°41.8' | 4152 | 3586 | 11/20/95 | 01/29/97 | 20 | |
| *CBmeso8 upper\** | 21°16.3' | 20°41.5' | 4120 | 745 | 01/30/97 | 06/04/98 | 8 | |
| **CBmeso9 lower** | 21°15.2' | 20°42.4' | 4121 | 3580 | 11/06/98 | 07/11/99 | 20 | |
| **CBmeso12 lower\*** | 21°16.0' | 20°46.5 | 4145 | 3610 | 04/05/01 | 12/17/01 | 14 | |
| **CBmeso13 lower** | 21°16.8' | 20°46.7' | 4131 | 3606 | 04/23/02 | 05/08/03 | 20 | |
| *CBmeso14 upper* | 21°17.2' | 20°47.6' | 4162 | 1246 | 05/31/03 | 04/18/04 | 10 | |
| **CBmeso15 lower** | 21°17.9' | 20°47.8' | 4162 | 3624 | 04/17/04 | 07/21/05 | 20 | |
| **CBmeso16 lower** | 21°16.8' | 20°47.8' | 4160 | 3633 | 07/25/05 | 09/28/06 | 20 | |
| **CBmeso17 lower** | 21°16.4' | 20.48.2' | 4152 | 3614 | 10/24/06 | 25.03.07 | 20 | |
| **CBmeso18 lower** | 21°16.9' | 20°48.1' | 4168 | 3629 | 03/25/07 | 04/05/08 | 20 | |
| **CBmeso19 lower** | 21°16.2' | 20°48.7' | 4155 | 3617 | 04/22/08 | 03/22/09 | 20 | |
| **CBmeso20 lower\*** | 21°15.6' | 20°50.7' | 4170 | 3620 | 04/03/09 | 05/25/09 | 4 | |

Asterisks represent traps with malfunctioning, leaving gaps in the diatom record.



Table 2: Species composition of the assemblage of diatoms at site CBmeso between March 1988 and
June 2009.

| Table 2 | | Romero et al. |
|---|---|---|
| Group | Species | Main References |
| 1) Benthic | | |
| | *Actinoptychus senarius* | Andrews (1981), Round et |
| | *Actinoptychus vulgaris* | al. (1990), Hasle and |
| | *Biddulphia alternans* | Syvertsen (1996). |
| | *Cocconeis* spp. | |
| | *Delphineis surirella* | |
| | *Diploneis* spp. | |
| | *Gomphonema* spp. | |
| | *Odontella mobiliensis* | |
| | *Paralia sulcata* | |
| | *Psammodyction panduriformis* | |
| | *Rhaphoneris amphiceros* | |
| | *Tryblionella* spp. | |
| 2) Coastal upwelling | | |
| | *Chaetoceros affinis* | Hasle and Syvertsen |
| | *Chaetoceros cinctus* | (1996); Abrantes et al. |
| | *Chaetoceros compresus* | (2002); Nave et al., 2001; |
| | *Chaetoceros constrictus* | Romero et al., 2002; |
| | *Chaetoceros coronatus* | Romero and Armand, |
| | *Chaetoceros debilis* | 2010) |
| | *Chaetoceros diadema* | |
| | *Chaetoceros radicans* | |
| | *Thalassionema nitzschioides* var. *nitzschioides* | |
| 3) Coastal planktonic | | |
| | *Actinocyclus curvatulus* | Romero and Armand |
| | *Actinocyclus octonarius* | (2010), Romero and |
| | *Actinocyclus subtilis* | Fischer (2017); Crosta et al. |
| | *Azpeitia barronii* | (2012); Romero et al. |
| | *Chaetoceros concavicornis* (vegetative cell) | (2009, 2012, 2020) |
| | *Coscinosdiscus argus* | |
| | *Coscinosdiscus centralis* | |
| | *Coscinosdiscus radiatus* | |
| | *Cyclotella litoralis* | |
| | *Proboscia alata* | |
| | *Shionodiscus oestrupii* var. *venrickae* | |
| | *Stellarima stellaris* | |
| | *Thalassionema pseudonitzschioides* | |
| | *Thalassiosira binata* | |
| | *Thalassiosira conferta* | |
| | *Thalassiosira delicatula* | |
| | *Thalassiosira dyporocyclus* | |
| | *Thalassiosira elsayedii* | |
| | *Thalssiosira poro-irregulata* | |





| | | |
|---|---|---|
| | *Thalassiosira rotula* | |
| 4) Open-ocean | | |
| | *Alveus marinus* | Romero and Armand |
| | *Aserolampra marylandica* | (2010), Romero and |
| | *Asteromphalus arachne* | Fischer (2017), Romero et |
| | *Asteromphalus cleveanus* | al. (2005), Crosta et al. |
| | *Asteromphalus flabellatus* | (2012), Romero et al. |
| | *Asteromphalus heptactis* | (2020). |
| | *Asteromphalus sarcophagus* | |
| | *Azpetia africana* | |
| | *Azpetia neocrenulata* | |
| | *Azpetia nodulifera* | |
| | *Azpetia tabularis* | |
| | *Bacteriastrum elongatum* | |
| | *Bogorovia* spp. | |
| | *Coscinodiscus reniformis* | |
| | *Detonula pumila* | |
| | *Fragilariopsis doliolus* | |
| | *Guinardia cyclindrus* | |
| | *Haslea* spp. | |
| | *Hemidiscus cuneiformis* | |
| | *Leptocyclindrus mediterraneus* | |
| | *Nitzschia aequatoriale* | |
| | *Nitzschia bicapitata* | |
| | *Nitzschia capuluspalae* | |
| | *Nitzschia interruptestriata* | |
| | *Nitzschia sicula* | |
| | *Nitzschia sicula* var. *rostrata* | |
| | *Planktoniella sol* | |
| | *Pseudo-nitzschia* spp. | |
| | *Pseudosolenia calcar-avis* | |
| | *Pseudotriceratium punctatum* | |
| | *Rhizosolenia acuminata* | |
| | *Rhizosolenia bergonii* | |
| | *Rhizosolenia imbricatae* | |
| | *Rhizosolenia robusta* | |
| | *Rhizosolenia setigera* | |
| | *Rhizosolenia styliformis* | |
| | *Roperia tessellata* | |
| | *Shionodiscus oestrupii* var. *oestrupii* | |
| | *Thalassionema bacillare* | |
| | *Thalassionema frauenfeldii* | |
| | *Thalassionema nitzschioides* var. *capitulata* | |
| | *Thalassionema nitzschioides* var. *inflata* | |
| | *Thalassionema nitzschioides* var. *lanceolata* | |
| | *Thalassionema nitzschioides* var. *parva* | |
| | *Thalassiosira eccentrica* | |
| | *Thalassiosira endoseriata* | |
| | *Thalassiosira ferelineata* | |
| | *Thalassiosira lentiginosa* | |
| | *Thalassiosira leptopus* | |
| | *Thalassiosira lineata* | |
| | *Thalassiosira nanolineata* | |





*Thalassiosira parthenia*
*Thalassiosira plicata*
*Thalassiosira punctigera*
*Thalassiosira sacketii* var. *sacketii*
*Thalassiosira sacketii* var. *plana*
*Thalassiosira subtilis*
*Thalassiosira symmetrica*
*Thalassiothrix* spp.





FIGURES

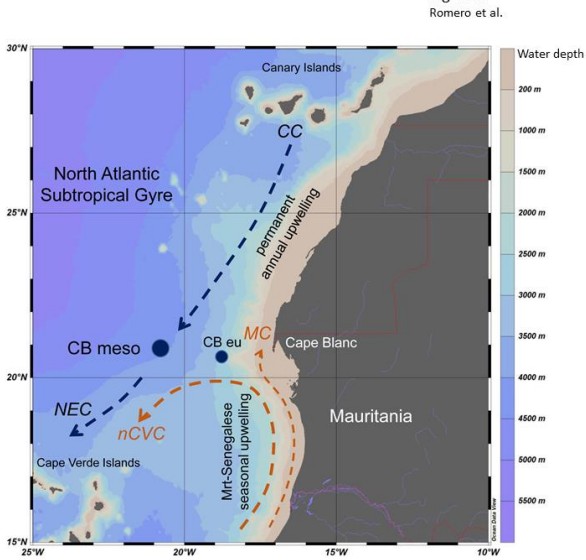

Figure 1. Map of the study area showing the locations of the sediment trap sites CBmeso and CBeu
        (dark blue dots). Major surface currents are also shown (Canary Current=CC; Mauritanian
        Current=MC; North Equatorial Current (NEC), north Cape Verde Current=nCVC). The upwelling
zones along the northwestern African margin are depicted after Cropper et al. (2014). The color
        scale (right-hand side) refers to meters below surface water (0 m).





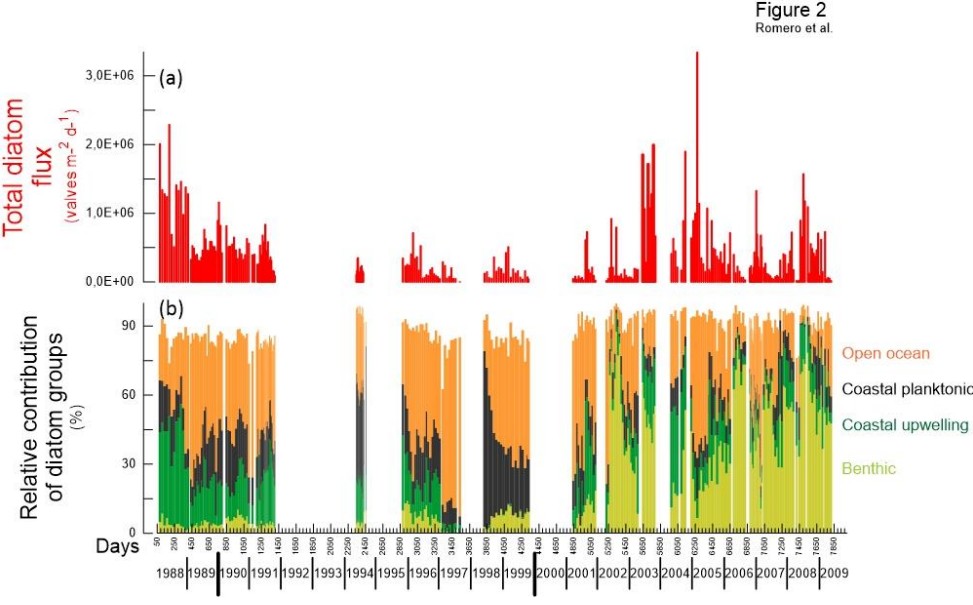

Figure 2. Total diatom flux (valves m$^{-2}$ d$^{-1}$) and relative contribution of diatom groups (relative

contribution, %) for the interval March 1998 and June 2009 at the CBmeso site. Groups of diatoms

are: benthic (light green), coastal planktonic (black), coastal upwelling (dark green), and open-

ocean (orange). For the species-specific composition of each group see 4.2. and Table 2. For

interpretation of the references to color in this figure legend, the reader is referred to the web

version of this article.





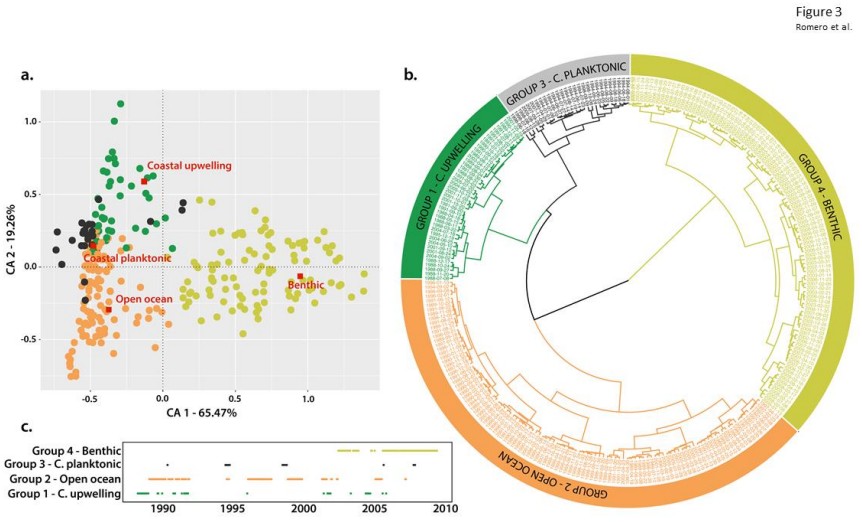

Figure 3. (a) Correspondence Analysis (CA) of diatom groups found at CBmeso site between March 1988 and June 2009, coupled with (b) a hierarchical clustering analysis of samples' score resulting from CA (see 2.3 Statistical analysis). Note that in 3a the red squares for each group represents the centroid of dates and their placement within the corresponding group. The corresponding group's name is written in red. (c) The time series of the four diatom groups identified by both multivariate analysis (CA and clustering) is also represented. Colours used for identifying each diatom group are the same as in Fig. 2b. Euclidean distance and Ward's aggregation link were used to perform the hierarchical dendrogram. For the species-specific composition of each group see 4.2. and Table 2. For interpretation of the references to colour in this figure legend, the reader is referred to the web version of this article.





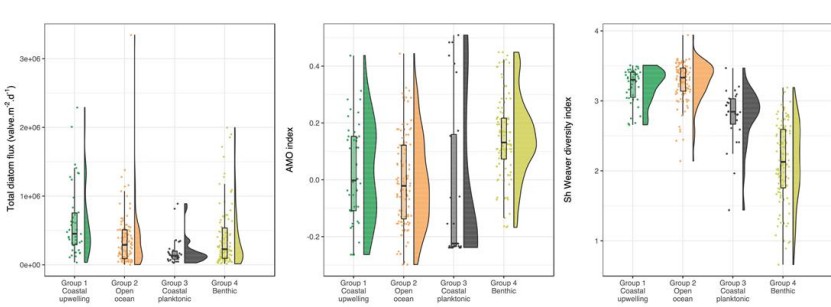

Figure 4: Comparison of (a) clusters extracted from multivariate analysis with the environmental

forcing variables (a1: Total diatom flux; a2: AMO; a3: Shannon diversity). Colours used for

identifying each diatom group are the same as in Fig. 2b.



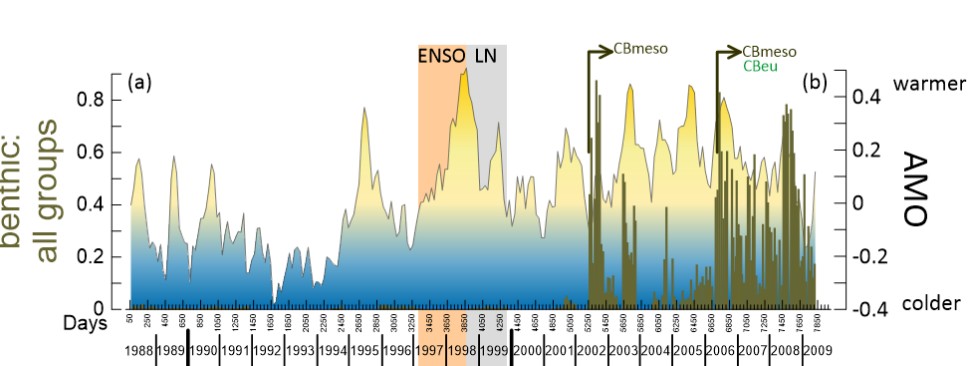

Figure 5: Time-series of ratio benthic:all groups (olive bars, a) at site CBmeso and the Atlantic

Multidecadal Oscillation (AMO, b) between March 1988 and May 2009. The fill in (b) represents the

colder phase (blue) and the warmer phase (yellow) of AMO. Inverted arrows in the lower panel

below the benthic:all groups bars represent the abrupt increase of relative contribution of benthic

diatoms, first seen at CBmeso in early winter 2002 and in winter 2006 at CBmeso and at CBeu

(Romero and Fischer, 2017; Romero et al., 2020). Shadings in the background: light orange, El

Niño/Southern Oscillation (ENSO); grey, La Niña (LN). For interpretation of the references to color

in this figure legend, the reader is referred to the web version of this article.





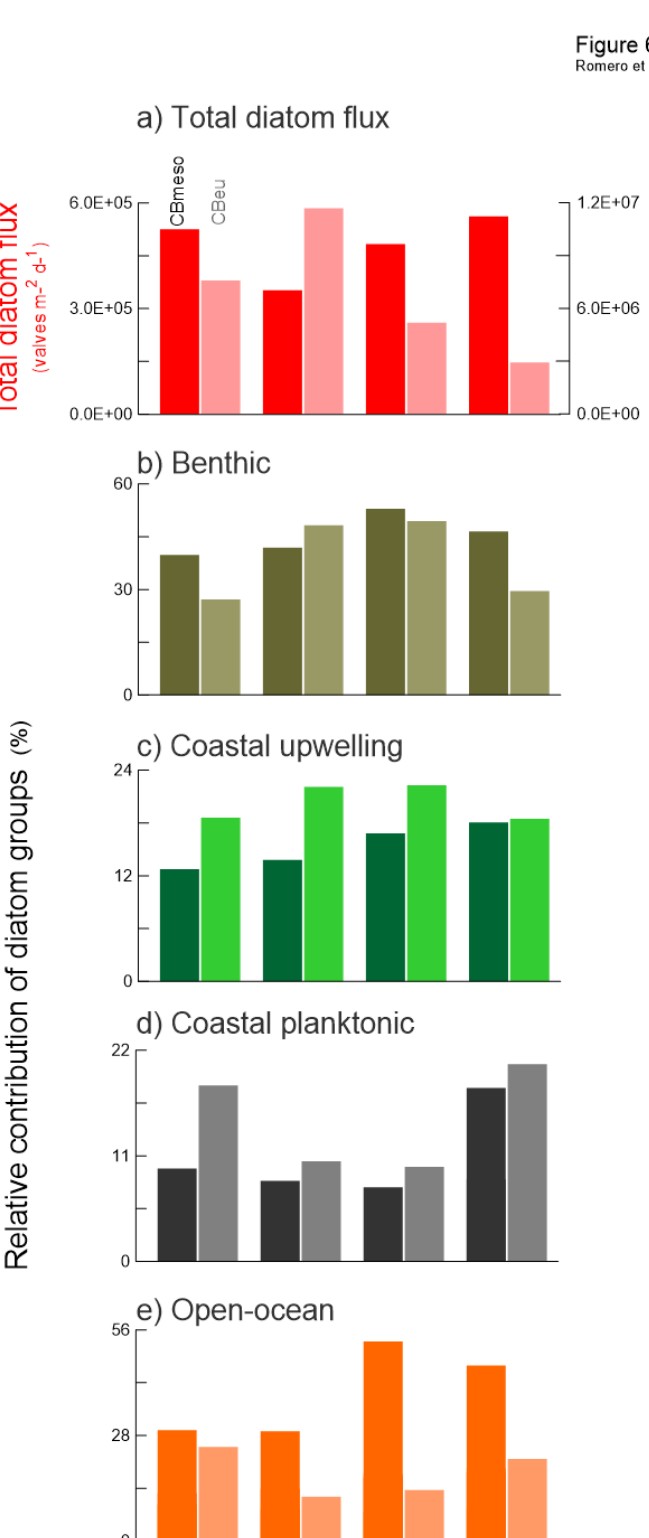



Figure 6: Comparison of seasonal values of (a) total diatom flux (valves m$^{-2}$ d$^{-1}$) and (b-e) the relative contribution of diatom groups (%) at sites CBmeso and CBeu (see 2.1 for trap locations). Darker

colors represent flux and relative percentage at CBmeso, while lighter those of CBeu. Note that the right y-axis for (a) total diatom flux correspond to CBeu and the left-hand y-axis to the CBmeso site. For the species-specific composition of each group at CBmeso see 4.2. and Table 2. The species-

specific composition of groups at CBeu is originally published in Romero and Fischer (2017). Colours used for identifying each diatom group are the same as in Fig. 2b. For interpretation of the references to color in this figure legend, the reader is referred to the web version of this article.