# Peer review of "A two-decades (1988-2009) record of diatom fluxes in the Mauritanian coastal upwelling: Impact of low-frequency forcing and a two-step shift in the species composition"

_Biogeosciences, 2020_

## Referee Comment (RC1) · Anonymous Referee #1 · 30 Oct 2020

The authors present an almost 20-year-long record of diatom fluxes retrieved from the Mauritanian coastal upwelling, a region of high productivity (and therefore economically important region). Authors performed an exhaustive taxonomical analysis of 282 samples, which represents a tremendous effort due to the high diversity of diatom assemblages collected by the traps. The authors grouped diatom assemblages in ecological groups and relate their temporal changes to different environmental parameters with particular emphasis in their relationship with climatic phenomena with multiannual variability such as the Atlantic Multidecadal Oscillation and El Niño–Southern Oscillation

(ENSO). Authors identified three major intervals throughout the time series based on the changes in the intensity and composition of the diatom fluxes. These intervals are linked with the previously mentioned multiannual climatic phenomena. It is important to highlight that this record probably represents the longest diatom-flux sediment trap ever reported in the global ocean, and that allowed to assess the interannual variability of diatom fluxes in a high productivity region (EBUES). The manuscript is well-written and novel and has high scientific impact. Moreover, the the statistical analysis appropriate and the figures of high quality. There are only a few aspects that require further clarification and discussion before the manuscript can be published in Biogeosciences. Find a list of my comments/suggestions below.

Line 25: Please include AMO between brackets the first time it is mentioned in the text.

Line 91: I believe authors could go a little bit further and state that this is the longest diatom time series sediment trap record of the world's ocean.

Line 108: Authors should be more specific and specify the depth range of the position of the sediment traps during their study in the text (i.e. not only in Table 1).

Material and methods Line 105 Since there are several gaps in the sediment trap record, authors could provide the number of days sampled during the 19-year record (i.e. the proportion of days sampled versus the total number of days). This would help the reader to have a better idea of the gaps in the record.

Line 108 Could authors provide sampling intervals for each deployment in Table 1 and at least a range in the main text?

Line 111. While I agree with this statement, there were two mooring deployments with sediment traps deployed at 700 m and therefore their collection efficiency could have been compromised. Since the collection interval of one of these deployments coincided with an strong ENSO event, it is important that authors discuss in the text the possibility of collection efficiency issues during these intervals.

Line 156. Could authors provide annual diatom valve estimates for the years with the most complete records? Even a rough estimate of the annual fluxes at this site would be useful for the specialized reader in order to be able to compare the diatom fluxes of this site with other regions of the global ocean.

Results 257 Could authors provide a rough estimation, i.e. average daily and/or annual fluxes, for radiolarian and silicoflagellates fluxes? This would help the reader to understand the contribution of both groups in relation to diatoms. Also, as mentioned before, I would suggest to provide annual estimates in order to facilitate the comparison of the diatom valve fluxes of this site with other regions of the global ocean.

259 Please avoid the use of acronym TDF, i.e. write the name in full.

266 "the highest"

367 "concluded"

Line 385 Please specify/repeat when this change occurs here.

Line 429 "5.1.2 The occurrence of the strong 1997 ENSO and the response of the diatom community off Mauritania" The intense ENSO event registered by the traps coincides with the use of sediment trap record collected at substantially shallower depth than most of the other deployments. According to Table 1 the sediment trap from deployment "CBmeso8 upper" was placed at around 700 m while most of the traps used in the experiment were placed at > 3000 m (with some exceptions). The collection area of the shallower sediment trap and collection efficiency of the "CBmeso8 upper" could be different than the other records, and therefore it could have affected the composition of the diatom assemblage collected during this interval. Authors should discuss this point in the text.

Line 455 Authors could also cite the possible impact of strong ENSO events on the Mediterranean diatom fluxes as reported by Bàrcena et al. (2004) and Rigual-Hernández et al. (2013).

Figures Figure 4. The graphs in this figure are too small for proper visualization. Please increase the size of the graphs.

References Bárcena, M.A., Flores, J.A., Sierro, F.J., Pérez-Folgado, M., Fabres, J. and Calafat, A., 2004. Planktonic response to main oceanographic changes in the Alboran Sea (Western Mediterranean) as documented in sediment traps and surface sediments. Marine Micropaleontology, 53(3-4): 375-398. Rigual-Hernández, A.S., Bárcena, M.A., Jordan, R.W., Sierro, F.J., Flores, J.A., Meier, K.J.S., Beaufort, L. and Heussner, S., 2013. Diatom fluxes in the NW Mediterranean: evidence from a 12-year sediment trap record and surficial sediments. Journal of Plankton Research, 35(5): 1109-1125.

---

## Referee Comment (RC2) · Anonymous Referee #2 · 16 Nov 2020

This paper by Romero, Ramondenc and Fischer provides a quasi 20yr diatom record collected by a sediment trap deployed in a mesotrophic region under the direct influence of the Cape Blanc filament. This new and important mesotrophic diatom dataset (CB meso) is compared with the equivalent record found in the trap with a more coastal location and consequently stronger influence by coastal upwelling (CB eu). Furthermore, the authors compare the record to large scale and low-frequency modes of climate and/or ocean circulation, and consider the variability encountered on the diatoms assemblage as a reflex of changes in the Canary Upwelling system at the Latitude

of Cape Blanc, which in turn is attributed to changes in the variability of global scale circulation mode reflected by the AMO index.

The paper is well structured and written and the results important, however I believe it could benefit from considering a few aspects mentioned below.

(Frankcombe et al., 2010) with the presented dataset the authors can only check on how major shifts between positive and negative states of AMO, occurred within the period of this record, affect the Canary Upwelling system, but not its fully and long-term effect on the system. However, the NAO index of atmospheric circulation over Europe has a periodicity in the order of 7-8 years (Knut Lehre Seip et al., 2019) and the work of (Yamamoto and Palter, 2016) shows a clear relation between the NAO and the AMO, with northerly winds associated to a positive state of AMO and zonal winds to a negative state of AMO. As such, it would be interesting to verify the relation of your data with NAO variability, since upwelling is indeed a response to an atmospheric process. It would also have been nice to have a comparison with the upwelling index or northerly wind strength. Maybe through another statistical approach, something like cross-correlation?

On which respects the effect of warming climate on the upwelling system and its primary production, you depart from the different conclusions reached by different studies, as presented in your introduction, to the proposal that your data is a different way of approaching the question. However, you conclude that your diatom data might be instrumental in distinguishing between climate-forced and intrinsic variability of the population of primary producers.

I have trouble with this statement, intrinsic variability is related to the basic needs of the organisms, so they will most probably change in function of the changes imposed on the system both by global and regional processes that in the end will also react to climate forcing!

Furthermore, although it is very important to understand the process behind your stunning increase in benthic diatoms, your record does not allow you to verify what happens in terms of the plankton production and assemblage evolution during this 20yr. Or does it? Can you deduce the effect of the benthic flux that obscures the total record, and explore the 20yr variability of the planktonic diatom flux and assemblages 'composition that reach the trap?

There is a general problem with the way references are listed in the text they do not follow an alphabetical order nor the year of publication.

What is the reason to use the term pelagial rather than pelagic? Although used for lakes I have not seen any paper that defends/justifies its use for the ocean environment.

The use of satellite images (composites for the n° of years considered for each specific time interval / diatom phase) for comparison would also be important to verify the variability on the surface water and upwelling conditions.

Different depths of trap deployment at some time intervals (Table 1) may influence the diatom assemblage encountered as a result of a different catching area and the different contribution of particles transported by intermediate nepheloid layers. This needs to be acknowledged and discussed especially because one of the periods coincides with the ENSO period.

Are you assuming that the intensification of the shelf and slope poleward current favors an increase in production of the benthic community and maintenance of the means of downslope transport, or the existence of a stronger poleward current gives rise to a stronger suspension of the benthic forms and their downslope transport in higher quantities? This needs clarification and discussion.

Specific notes: Pg. 3, Ln. 78 – The authors suggest that a different approach for the characterization of multiyear to interdecadal upwelling intensity in EBUEs is by assessing fluxes of particulates and microorganisms as captured by continuous sediment trap experiments."

Although you can assume that the flux of planktonic organism blooming in surface waters as a result of upwelling intensity, we are also aware that the nutrient content of the upwelling water is determinant for the size of the blooms as well as for the type of phytoplankton community. As such, bloom size and consequently microorganism fluxes could also reflect shifts in the upwelling source water associated with latitudinal shifts for example, rather than variations in upwelling intensity.

In fact, in this study besides the physical setting it is important to also consider the chemical (nutrient) and biological setting.

Pg. 6, Ln. 103 – The SACW occurs in layers between 100 and 400 m depth at the Banc d'Arguin and off Mauritania.

Pg. 8, Ln 250-252 – ENSO appears to be modulated by AMO, check Levin et al, (2017) or Chen et al., 2019 or Zhang et al., (2019).

Pg. 9. Ln. 301 – 302 – The list of species presented do correspond to marine plankton forms that although not thriving in the highly productive and colder coastal upwelling systems, and more likely to be found in warmer waters, they are also not characteristic or real oligotrophic waters.

Pg. 10, Ln. 323- 324 - The impact of the environmental variables on diatom communities was investigated by simple comparison using the samples clustering and the forcing values associated (Fig 4).

You are not using the forcing values, but rather the value of an index that is considered to define the coherent mode of natural variability occurring in the north Atlantic. Changes in this mode will have an impact on the circulation at your study site and be considered a forcing factor for your specific process.

Pg. 10, Ln. 329 – Please specify tendency of the gradient.

Pg. 10, Ln. 335 – Mentioned figure should be included as a supplementary figure.
Pg. 10, Ln 337 - the benthic diatom D. surirella decreased the diversity, although it also seems to be promoted determined by AMO strengthening. In the same way, the second CA axis samples scores are positively correlated with TDF, which confirms that coastal upwelling diatoms seems to promote define the TDF.

Pg.11, Ln 352 - Based on outstanding shifts in the species-specific composition of the diatom assemblage occurred throughout the study interval (Fig. 2b).

Please reformulate.

Pg. , Ln. 360 - Based on the long-term trends of our data and their statistical analysis (Figs. 2-5), we suggest that the proposed intervals were the response of the diatom populations to the impact of low frequency forcing on the Canary upwelling system.

To be correct, the upwelling system is the one that responds to the low frequency forcing. Diatom assemblages reflect hydrographic and nutrient availability brought up by the upwelled source waters.

Figure 4: Comparison of (a) clusters extracted from multivariate analysis with the environmental forcing variables (a1: Total diatom flux; a2: AMO; a3: Shannon diversity).

Besides being too small and difficult to see, total diatom Flux and Diversity are not forcing variables. They all reflect the community adaptation to the regional conditions resulting from the forcing factor(s).

References

Chen, S., Song, L. & Chen, W.: Interdecadal Modulation of AMO on the Winter North Pacific Oscillation-Following Winter ENSO Relationship. Adv. Atmos. Sci. 36, 1393–1403, 2019. https://doi.org/10.1007/s00376-019-9090-1

Frankcombe, L. M., Heydt, A. v. d., and Dijkstra, H. A.: North Atlantic Multidecadal Climate Variability: An Investigation of Dominant Time Scales and Processes, Journal of Climate, 23, 3616-3638, 2010.

[Figure]

Levine, A. F. Z., M. J. McPhaden, and D. M. W.Frierson: The impact of the AMO on multidecadal ENSO variability, Geophys. Res. Lett., 44, 3877–3886, 2017. doi:10.1002/2017GL072524.

Knut Lehre Seip, Øyvind Grøn and Hui Wang: The North Atlantic Oscillations: Cycle Times for the NAO, the AMO and the AMOC. Climate, 2019, 7, 43; doi:10.3390/cli7030043

Yamamoto, A. and Palter, J. B.: The absence of an Atlantic imprint on the multidecadal variability of wintertime European temperature, Nature Communications, 7, 10930, 2016.

Zhang, W., X. Mei, X. Geng, A. G. Turner, and F. Jin: A Nonstationary ENSO–NAO Relationship Due to AMO Modulation. J. Climate, 32, 33–43, 2019. https://doi.org/10.1175/JCLI-D-18-0365.1.

---

## Author Comment (AC1) · 14 Dec 2020

A two-decades (1988-2009) record of diatom fluxes in the Mauritanian coastal upwelling: Impact of low-frequency forcing and a two-step shift in the species composition composition (bg-2020-336) Authors = Oscar E. Romero, Simon Ramondenc and Gerhard Fischer

Response to Referee 1's comments

As required by BG, the response to the Referees is structured in the following sequence: (1) comments from Referee 1 (RC1) and (2) authors' comments (AC).

Comments from Referee #1

RC1: Line 25: Please include AMO between brackets the first time it is mentioned in the text.

AC: This will be accordingly corrected.

RC1: Line 91: I believe authors could go a little bit further and state that this is the longest diatom time series sediment trap record of the world's ocean.

AC: the sentence will be accordingly rephrased, and we will mention that the presented sediment trap-based diatom record is the longest so far known.

RC1: Material and methods Line 105 Since there are several gaps in the sediment trap record, authors could provide the number of days sampled during the 19-year record (i.e. the proportion of days sampled versus the total number of days). This would help the reader to have a better idea of the gaps in the record.

AC: the entire study interval (March 1988 until May 2009) extended over 7,734 days. Out of them, samples were collected for 5574 days. The gaps totalize 2160 days. This will be mentioned in Materials and Methods.

RC1: Line 108: Authors should be more specific and specify the depth range of the position of the sediment traps during their study in the text (i.e., not only in Table 1).

AC: the depth range of the traps will be included in the text of the revised version. The range varies as follows: CB1lower, 2,195 m; CB2-5, 7, 9-12, 15-20lower: 3,502-3,633 m, and CB6, 8 and 14upper: 745-1,246 m.

RC1: Line 111. While I agree with this statement, there were two mooring deployments with sediment traps deployed at 700 m and therefore their collection efficiency could have been compromised. Since the collection interval of one of these deployments coincided with an strong ENSO event, it is important that authors discuss in the text

the possibility of collection efficiency issues during these intervals.

AC: It is true that the sediment trap CB8upper, which corresponds to the 1997-early 98 ENSO interval, was deployed at 745 m, while CB9lower was deployed at 3,580 m depth. However, the composition of the diatom assemblage (relative abundance) with the highest contribution of open-ocean and coastal planktonic diatoms -indicative of moderate to low nutrient conditions- shows a significant match between both traps. This is independent of the flux numbers. The total diatom flux is very low in both traps as well, with hardly any dramatic increase or decrease with depth. This is interpreted as sound evidence of no significant difference in the collection efficiency of both traps (despite different trap deployment depths).

RC1: Line 156. Could authors provide annual diatom valve estimates for the years with the most complete records? Even a rough estimate of the annual fluxes at this site would be useful for the specialized reader in order to be able to compare the diatom fluxes of this site with other regions of the global ocean.

AC: this is a helpful suggestion of R1. Yearly fluxes deliver a broader picture of interannual variations at the CBmeso location, as representative of offshore migration of the chlorophyll filament, productivity variations and upwelling intensity, but they are also helpful to compare the CBmeso location with fluxes from other trap locations, deployed either in similar or different oceanographic settings. We will present a Table with yearly fluxes of total diatoms for 13 calendar years and the results will be discussed in 5.1.

RC1: Results - Could authors provide a rough estimation, i.e. average daily and/or annual fluxes, for radiolarian and silicoflagellates fluxes? This would help the reader to understand the contribution of both groups in relation to diatoms. Also, as mentioned before, I would suggest to provide annual estimates in order to facilitate the comparison of the diatom valve fluxes of this site with other regions of the global ocean.

AC: although we agree that fluxes of silicoflagellates and radiolarians can be of interest for scientists working on marine siliceous plankton, we emphasize that the focus of our

long-term traps record is on the diatom fluxes and the species-specific composition of the assemblage. We believe that including data on silicoflagellates and radiolarians would be beyond the MS' focus and would lead the discussion in a quite different direction. There is also a methodological aspect to consider: the use of permanent slides for diatom and silicoflagellates census does not allow the proper quantification of radiolarian skeletons. Due to the low volume used in the preparation of the permanent slides, the low absolute concentration of skeletons is too low for reliable radiolarian census.

RC1: Please avoid the use of acronym TDF, i.e. write the name in full.

AC: The acronym will be avoided throughout the revised version of our MS.

RC1: 266 "the highest"

AC: The sentence has been rephrased and reads now: "Spring and summer show the highest amount of above-the-average total diatom concentration."

RC1: 367 "concluded"

AC: the sentence has been corrected to: "Using observational data and model experiments, Wang and Zhang (2013) concluded. . ."

RC1: Line 385 Please specify/repeat when this change occurs here.

AC: the corrected sentence reads now: "An extraordinary feature of the multiyear dynamics of diatom populations at the CBmeso site is the sharp shift in the species contribution in Mai 2001 (Fig. 2b)."

RC1: Line 429 "5.1.2 The occurrence of the strong 1997 ENSO and the response of the diatom community off Mauritania" The intense ENSO event registered by the traps coincides with the use of sediment trap record collected at substantially shallower depth than most of the other deployments. According to Table 1 the sediment trap from deployment "CBmeso8 upper" was placed at around 700 m while most of the traps used

in the experiment were placed at > 3000 m (with some exceptions). The collection area of the shallower sediment trap and collection efficiency of the "CBmeso8 upper" could be different than the other records, and therefore it could have affected the composition of the diatom assemblage collected during this interval. Authors should discuss this point in the text.

AC: It is true that the trap CB8upper was shallower than CB9lower (745 m vs 3,580 m trap depth, respectively; Table 1 in manuscript, see comment above about this issue). However, as mentioned above, we believe that the strong resemblance in the species-specific composition of the diatom assemblage of both traps (highest contribution of diatoms associated with waters of moderate to low nutrient content), without any significant percentage shift, delivers sound evidence on the reliability of the diatom data at CB8 and CB9. We will comment this issue in the revised version of our MS and include this sentence: "We are aware that the two traps temporally corresponding to 1997-1999 ENSO and La Niña were deployed at different depths. Although this might have impacted on the total diatom flux (the lower the trap, stronger the dissolution effect), the good match in the species-specific composition of the diatom assemblage at CB8 and CB9 traps points to the reliable signal of environmental response of the diatoms to an ENSO impact."

RC1: Line 455 Authors could also cite the possible impact of strong ENSO events on the Mediterranean diatom fluxes as reported by Bárcena et al. (2004) and Rigual-Hernández et al. (2013).

AC: We are grateful to Referee 1 for reminding us of these two papers on the ENSO impact on the phytoplankton dynamics in the western Mediterranean Sea. Both articles will be mentioned and discussed in the revised version.

RC1: Figure 4. The graphs in this figure are too small for proper visualization. Please increase the size of the graphs.

AC: the revised version of our MS will include a larger and better resolved file of Figure

4.

---

## Author Comment (AC2) · 14 Dec 2020

A two-decades (1988-2009) record of diatom fluxes in the Mauritanian coastal up-welling: Impact of low-frequency forcing and a two-step shift in the species composition (bg-2020-336) Authors = Oscar E. Romero, Simon Ramondenc and Gerhard Fischer

Response to Referee 2's comments

As required by BG, the response to the Referees is structured in the following sequence: (1) comments from Referee 2 (RC2) and (2) authors' comments (AC).

Comments from Referee #2 RC2: (Frankcombe et al., 2010) with the presented dataset the authors can only check on how major shifts between positive and negative states of AMO, occurred within the period of this record, affect the Canary Upwelling system, but not its fully and longterm effect on the system. However, the NAO index of atmospheric circulation over Europe has a periodicity in the order of 7-8 years (Knut Lehre Seip et al., 2019) and the work of (Yamamoto and Palter, 2016) shows a clear relation between the NAO and the AMO, with northerly winds associated to a positive state of AMO and zonal winds to a negative state of AMO. As such, it would be interesting to verify the relation of your data with NAO variability, since upwelling is indeed a response to an atmospheric process. It would also have been nice to have a comparison with the upwelling index or northerly wind strength. Maybe through another statistical approach, something like cross-correlation?

AC: We provide an additional analytical test that supports our interpretation. Indeed, we performed a correlation analysis with samples' score resulting from CA (Dim.1, Dim.2 and Dim. 3, which discriminates the diatom communities), climatic indexes (ENSO, NAO, AMO), diversity index (Shannon diversity) and fluxes (total diatom flux, freshwater diatom flux, Opal flux). As suggested by Reviewer 2, the correlogram shows a significant negative relationship between AMO and NAO. However, the goodness of fit between climatic indexes was low (R2 around 0.2). The correlogram also shows that the samples' score of first CA axis (Dim. 1, which discriminates the benthic from the other diatom groups) seems also impacted by the NAO, although with an exceptionally low R2. However, the statistical tests (clustering, boxplot and the Kruskall Wallis approach) performed in the first submission do not show any relationship between diatom groups and the NAO. Conversely to the correlogram, our statistical approach analyses each community independently and does not compare one group with the others. Although both statistical approaches are correct, we believe that the correlogram method could induce some misunderstanding, leading to a certain degree of overestimation of NAO impact. We conclude that AMO have a stronger impact on diatom communities off Mauritania than NAO. The significant NAO impact observed in correlogram is indirectly linked to communities via AMO, which is himself impacted by the NAO. We will comment on the possible impact of NAO on the diatom community at site CBmeso and discuss the publications suggested by Referee 2.

RC2: On which respects the effect of warming climate on the upwelling system and its primary production, you depart from the different conclusions reached by different studies, as presented in your introduction, to the proposal that your data is a different way of approaching the question. However, you conclude that your diatom data might be instrumental in distinguishing between climate-forced and intrinsic variability of the population of primary producers. I have trouble with this statement, intrinsic variability is related to the basic needs of the organisms, so they will most probably change in function of the changes imposed on the system both by global and regional processes that in the end will also react to climate forcing!

AC: indeed, this sentence is not as clear as we thought it was. We will rephrase it as follows: Our 1988-2009 data set contributes to distinguish the impact of low-frequency climate forcings and will be especially helpful for establishing the scientific basis for forecasting and modelling future states of the Canary EBUE and its decadal changes.

RC2: Furthermore, although it is very important to understand the process behind your stunning increase in benthic diatoms, your record does not allow you to verify what happens in terms of the plankton production and assemblage evolution during this 20yr. Or does it? Can you deduce the effect of the benthic flux that obscures the total record, and explore the 20yr variability of the planktonic diatom flux and assemblages 'composition that reach the trap?

AC: It is true that the dramatic shift in the species-specific composition of the diatom assemblage in May 2001 does not imply any dramatic change in the absolute values of the total diatom concentration nor it translated into any significant changes of the biogenic silica (=opal) flux (see also Romero et al., 2017, Prog. Oceanogr. 159, 131). This observation also matches previous work at CBmeso (Fischer et al., 2016, Biogeosciences 13, 3203).

RC2: There is a general problem with the way references are listed in the text they do not follow an alphabetical order nor the year of publication.

AC: The citation of articles and book chapters follow BG Instructions to Authors. We will check all quoted references again before re-submission.

RC2: What is the reason to use the term pelagial rather than pelagic? Although used for lakes I have not seen any paper that defends/justifies its use for the ocean environment.

AC: we will rephrase accordingly and use pelagic instead of pelagial throughout the MS.

RC2: The use of satellite images (composites for the n_ of years considered for each specific time interval / diatom phase) for comparison would also be important to verify the variability on the surface water and upwelling conditions.

AC: three pictures of chlorophyll a concentration have been added to Figure 1 (attached). They depict the average concentration of chlorophyll a for three winters (1997, 2002 and 2008), gained with SeaWIFs and MODIS (https://oceancolor.gsfc.nasa.gov/cgi/l3, details will be provided in the revised version of the MS). The high interannual variability is clearly recognized.

RC2: Different depths of trap deployment at some time intervals (Table 1) may influence the diatom assemblage encountered as a result of a different catching area and the different contribution of particles transported by intermediate nepheloid layers. This needs to be acknowledged and discussed especially because one of the periods coincides with the ENSO period.

AC: This issue was also raised by R1 and is addressed in the replies to Referee 1's comments.

RC2: Are you assuming that the intensification of the shelf and slope poleward current favors an increase in production of the benthic community and maintenance of the means of downslope transport, or the existence of a stronger poleward current gives rise to a stronger suspension of the benthic forms and their downslope transport in higher quantities? This needs clarification and discussion.

AC: The benthic diatoms found in the CBmeso trap samples predominantly thrive in shallow coastal waters, not deeper than 50 m water depth, within the sunlit layer along the Mauritanian coast. Its occurrence in the hemipelagic CBmeso trap represents a lateral transport signal. As a possible explanation for the dramatic increase of benthic diatoms in the hemipelagic environment, we speculate that the intensification of the shelf and slope poleward transport upon deeper waters. It is well-known that the dynamic Mauritanian coastal waters serve as a jet for cross-shore particle transfer and it produces sporadic particle clouds, which are advected hundred kilometres offshore within intermediate and bottom-near nepheloid layers (Nowald et al., 2015) toward the hemipelagic of the low-latitude NE Atlantic (Fischer and Karakaş, 2009). This transport occurs within weeks (Karakaş et al., 2006, 2009). Subsurface waters (200 to 300 m depth) might be the place of mixing processes of older, laterally-advected materials from the shelf by giant filament activity, with relatively fresh material derived from the open ocean surface (Fischer et al., 2009). In addition to the nepheloid layer-mediated transport, the dynamics of water masses related to the existence of the canyon system off Mauritania might have contributed to the enhancement of transport from shallow water upon the trap site CBmeso.

RC2: Pg. 3, Ln. 78 – The authors suggest that a different approach for the characterization of multiyear to interdecadal upwelling intensity in EBUEs is by assessing fluxes of particulates and microorganisms as captured by continuous sediment trap experiments."

AC: As stated in the paragraph of our first submitted version (l. 66-77), a vast majority of the previous studies on the long-term variability of productivity and upwelling intensity along the north-western African margin follows different approaches than the one applied in our study. Approaches previously used for the description of interannual upwelling variations are velocity and directions of winds, annual wind stress, and Ekman transport. With the sentence written in last paragraph of the Introduction, we intended to emphasize that observational data based on interannual trap experiments are uncommon and represent a different approach to the study of possible links between variability of the microorganisms community, upwelling variations and the impact of low-impact climate and oceanographic forcing.

RC2: Although you can assume that the flux of planktonic organism blooming in surface waters as a result of upwelling intensity, we are also aware that the nutrient content of the upwelling water is determinant for the size of the blooms as well as for the type of phytoplankton community. As such, bloom size and consequently microorganism fluxes could also reflect shifts in the upwelling source water associated with latitudinal shifts for example, rather than variations in upwelling intensity. In fact, in this study besides the physical setting it is important to also consider the chemical (nutrient) and biological setting.

AC: it is true that the occurrence of diatom populations (or those of any other organisms) at the CBmeso site is the result of the interaction of several processes acting in different timescales. The fact that the shift in the species-specific composition of the diatom assemblage in May 2001 is not paralleled by either an increase or decrease of total diatom and/or biogenic silica flux suggests that the intensity of upwelling per se did not significantly change after 2001, nor an increase in DSi availability occurred after May 2001 in waters overlying site CBmeso.

RC2: Pg. 6, Ln. 103 – The SACW occurs in layers between 100 and 400 m depth at the Banc d'Arguin and off Mauritania.

AC: the sentence will be accordingly corrected.

RC2: Pg. 8, Ln 250-252 – ENSO appears to be modulated by AMO, check Levin et al,

(2017) or Chen et al., 2019 or Zhang et al., (2019).

AC: these three papers will be discussed in the revised version.

RC2: Pg. 9. Ln. 301 – 302 – The list of species presented do correspond to marine plankton forms that although not thriving in the highly productive and colder coastal upwelling systems, and more likely to be found in warmer waters, they are also not characteristic or real oligotrophic waters.

AC: in addition to other peer-reviewed publications, we base the grouping of diatom species found in the CBmeso trap samples on our almost 20 year continuous research of the temporal dynamics of diatom populations and their biogeographical occurrence. Throughout the years, we have learnt that the species listed as 'open-ocean taxa' are typical of ocean waters of low content of dissolved silica (DSi). The term 'low' here is sued in comparison to the high content of DSi in coastal waters of EBUEs, which is at least five to ten times higher than in open-ocean waters of the mid-latitude North Atlantic. From this point of view, we are confident in characterizing the open-ocean diatoms (as listed in Table 2 of our MS) as typical of oligotrophic waters. In addition to the papers quoted in out MS, we have studied trap-gained diatom assemblages from different setting from mid- and low latitudes all around the world. These studies have largely helped to characterize the biogeography of marine diatoms and their ecology (e.g., Romero et al., 2000, Deep Sea Research Part II: 47(9): 1939; Romero et al., 2002, Journal of Plankton Research 24: 1035; Romero et al., 2009, Deep Sea Research I 56: 571; Romero and Armand, 2010 in: The Diatoms, Applications for the Environmental and Earth Sciences, Ed.: Smol and Stoermer, Cambridge University Press: 373; Romero et al., 2016, Progress in Oceanography 147, 38).

RC2: Pg. 10, Ln. 323- 324 - The impact of the environmental variables on diatom communities was investigated by simple comparison using the samples clustering and the forcing values associated (Fig 4). You are not using the forcing values, but rather the value of an index that is considered to define the coherent mode of natural variability occurring in the north Atlantic. Changes in this mode will have an impact on the circulation at your study site and be considered a forcing factor for your specific process.

AC: We agree with Referee 2 in that we used climate indexes, which is a proxy of the direct environmental forcing. We did not use highly-resolved environmental data (e.g., DSi content) because they are not available for the complete time serie.

RC2: Pg. 10, Ln. 329 – Please specify tendency of the gradient.

AC: it has been re-phrased and reads as follows: In addition, a gradient in the Shannon diversity index of the diatom populations (Fig. 4c) is observed with predominant low values (1.7-2.5) corresponding to benthic (=group 4), intermediate values (2.7-3) for coastal planktonic (=group 3) and high values (3.1-3.45) in samples dominated by coastal upwelling and open-ocean populations (=groups 2/1) (pairwise Wilcoxon rank sum test; p-value<0.05).

RC2: Pg. 10, Ln. 335 – Mentioned figure should be included as a supplementary figure.

AC: Figure 3 highlights our statistical approach to define which diatom communities dominate our samples and the time series of their respective dominance instead of doing it visually. Since this figure is also causally related to Figure 4, we do believe that Figure 3 should be kept as part of the MS figures and does not need to be transferred to Supplement

RC2: Pg. 10, Ln 337 - the benthic diatom D. surirella decreased the diversity, although it also seems to be promoted determined by AMO strengthening. In the same way, the second CA axis samples scores are positively correlated with TDF, which confirms that coastal upwelling diatoms seems to promote define the TDF.

AC: it has been re-phrased and reads as follows: 'Given that the first CA is positively driven by the benthic group, this confirms the outstanding dominance of the benthic

diatom D. surirella after May 2001, which also appears linked to the strengthening of AMO. In the same way, the second CA axis is positively correlated with total diatom flux also confirms that coastal upwelling diatoms deliver a large numbers of diatom valves.'

RC2: Pg.11, Ln 352 - Based on outstanding shifts in the species-specific composition of the diatom assemblage occurred throughout the study interval (Fig. 2b).

AC: it has been re-phrased and reads now: 'Based on outstanding shifts in the species-specific composition of the diatom assemblage occurred throughout the study interval (Fig. 2b), we propose three main intervals in the multiyear evolution of populations and discuss them in view of mayor environmental forcings:...'

RC2: Pg. , Ln. 360 - Based on the long-term trends of our data and their statistical analysis (Figs. 2-5), we suggest that the proposed intervals were the response of the diatom populations to the impact of low frequency forcing on the Canary upwelling system. To be correct, the upwelling system is the one that responds to the low frequency forcing. Diatom assemblages reflect hydrographic and nutrient availability brought up by the upwelled source waters.

AC: It is true that the upwelling in the Canary EBUE responds to low-frequency climate impact. By studying the diatom populations, we did not, however, directly characterize long-term variability of upwelling intensity off Mauritania as studies quoted in the Introduction of our first submitted version did (L. 66 th 77). Therefore, we believe that the sentence as written is correct.

RC2: Figure 4: Comparison of (a) clusters extracted from multivariate analysis with the environmental forcing variables (a1: Total diatom flux; a2: AMO; a3: Shannon diversity). Besides being too small and difficult to see, total diatom Flux and Diversity are not forcing variables. They all reflect the community adaptation to the regional conditions resulting from the forcing factor(s).

AC: we will rephrase this accordingly. NAO, AMO, ENSO and the diversity index

Shannon-Weaver are indices while the total diatom flux is a variable. This was wrongly described in the original submission. The file resolution of Fig. 4 will be enlarged.

RC2: References Chen, S., Song, L. & Chen, W.: Interdecadal Modulation of AMO on the Winter North Pacific Oscillation-Following Winter ENSO Relationship. Adv. Atmos. Sci. 36, 1393– 1403, 2019. https://doi.org/10.1007/s00376-019-9090-1 Frankcombe, L. M., Heydt, A. v. d., and Dijkstra, H. A.: North Atlantic Multidecadal Climate Variability: An Investigation of Dominant Time Scales and Processes, Journal of Climate, 23, 3616-3638, 2010. Levine, A. F. Z., M. J. McPhaden, and D. M.W.Frierson: The impact of the AMO on multidecadal ENSO variability, Geophys. Res. Lett., 44, 3877–3886, 2017. doi:10.1002/ 2017GL072524. Knut Lehre Seip, Øyvind Grøn and Hui Wang: The North Atlantic Oscillations: Cycle Times for the NAO, the AMO and the AMOC. Climate, 2019, 7, 43; doi:10.3390/cli7030043 Yamamoto, A. and Palter, J. B.: The absence of an Atlantic imprint on the multidecadal variability of wintertime European temperature, Nature Communications, 7, 10930, 2016. Zhang, W., X. Mei, X. Geng, A. G. Turner, and F. Jin: A Nonstationary ENSO– NAO Relationship Due to AMO Modulation. J. Climate, 32, 33–43, 2019. https://doi.org/10.1175/JCLI-D-18-0365.1. AC: we are grateful for these references. These peer-reviewed publications will be accordingly discussed in the revised version.
* * *
[Figure]

Figure 1
Romero et al.

[Figure]

**Fig. 1.** Figure 1 revised

---

## Author Comment (AC3) · 4 Feb 2021

Dr. Oscar E. Romero
Senior Scientist
MARUM
D-28359 Bremen
Germany

Tel.      +49 421 218 – 65 645
E-Mail  oromero@uni-bremen.de
www    www.marum.de

Bremen, February 04th 2021

Associate Editor

*Biogeosciences*

Dr. Ny Riavo G. Voarintsoa

Subject: *comments by Referee 2 on revised version of MS bg-2020-336.*

.

Dear Dr. Voarintsoa:

We thank for the speed revision of our revised manuscript bg-2020-336. We are glad that Referee 1 agreed on the revised version.

Here, we comment on Referee 2's suggestions. We must admit that we are confused regarding the posted comments. As you certainly are able to check on our file uploaded on Jan 19th 2021 (file = *bg-2020-336-manuscript-version4.pdf*), the issues raised by Ref 2 in her/his review of Nov 16th 202 were (*i*) thoroughly addressed before submitting the revised version and (*ii*) responded, commented and/or discussed in the file '*Final reply to BGD Comments _ Rev 2_bg_2020_336*', also uploaded on Jan 19th 2021. Below we comment (*in red italics*) on each of the issues raised by Ref 2.

RC2: What is the reason to use the term pelagial rather than pelagic? Although used for lakes I have not seen any paper that defends/justifies its use for the ocean environment.
AC: we will rephrase accordingly and use pelagic instead of pelagial throughout the MS.
The term is still used in the new version that was submitted.
*New AC: we had replaced 'pelagial' for 'pelagic' throughout in the revised version uploaded on Jan 19th 2021.*

RC2: The use of satellite images (composites for the n_ of years considered for each specific time interval / diatom phase) for comparison would also be important to verify the variability on the surface water and upwelling conditions.
AC: three pictures of chlorophyll a concentration have been added to Figure 1 (attached). They depict the average concentration of chlorophyll a for three winters (1997, 2002 and 2008), gained with SeaWIFs and MODIS (https://oceancolor.gsfc.nasa.gov/cgi/l3, details will be provided in the revised version of the MS). The high interannual variability is clearly recognized.
The figure 1 associated to the response letter does not correspond to the figure in the uploaded version, that is still the figure of version 1
*New AC: the revised Figure 1 showed three satellite pictures (1b-c), which represent the average winter season of three years.*

RC2: Figure 4: Comparison of (a) clusters extracted from multivariate analysis with the environmental

forcing variables (a1: Total diatom flux; a2: AMO; a3: Shannon diversity). Besides being too small and difficult to see, total diatom Flux and Diversity are not forcing variables. They all reflect the community adaptation to the regional conditions resulting from the forcing factor(s).

AC: we will rephrase this accordingly. NAO, AMO, ENSO and the diversity index C9 BGD Interactive comment Printer-friendly version Discussion paper Shannon-Weaver are indices while the total diatom flux is a variable. This was wrongly described in the original submission. The file resolution of Fig. 4 will be enlarged.

Not really done. The figure has the same exact size!

*New AC: a larger (highly resolved) file for this figure was in the pdf file bg-2020-336-manuscript-version4.pdf. In case its resolution is not high enough, we will upload a larger file for the ready-to-print version.*

New Notes

352 - Based on outstanding shifts in the species-specific composition of the diatom assemblage occurred throughout the study interval (Fig. 2b). We propose three main intervals in the multiyear…. First sentence does not make sense. I believe the second sentence should follow without the full stop after Fig 2b.

*AC: this sentence reads in the revised MS file (l. 377-379): 'Based on outstanding shifts in the species-specific composition of the diatom assemblage occurred throughout the study (Figs. 2b and 3), we propose three main intervals in the multiyear evolution of populations and discuss them in view of mayor environmental forcings:…'*

395 - Fig. 4). The dominance of benthic taxa also prevails (after) afterward throughout until recently recovered traps at site CBmeso (Oscar E. Romero, unpublished data).

*AC: this does not correspond with the line numbering in the revised version.*

399 – 403 Paragraph needs rewriting

*AC: line 399 is the end of a paragraph and l. 400 – 403 contains the first two sentences of the next paragraph. These two sentences seem quite correct to us.*

424 – 428 Paragraph needs rewriting

*AC: The sentences at the beginning of this paragraph seem correct to us.*

We hope that these comments help you to clarify this confusion.

With best wishes,

Oscar E. Romero
and co-authors

---

## Author Response (AR3)

 Universität Bremen

Center for Marine Environmental Sciences
DFG Research Center · Cluster of Excellence
University of Bremen    www.marum.de

Dr. Oscar E. Romero
Senior Scientist
MARUM
D-28359 Bremen
Germany

Tel.    +49 421 218 – 65 645
E-Mail  oromero@uni-bremen.de
www    www.marum.de

Bremen, January 19th 2021

Associate Editor

*Biogeosciences*

Dr. Ny Riavo G. Voarintsoa

*Subject: revised version of MS bg-2020-336.*

.

Dear Dr. Voarintsoa:

We submit the revised version of your manuscript bg-2020-336. We have endeavored to deal with all the issues raised by both Anonymous Referees. Following both reviews, several changes were made to the text, figures, and tables. We have now uploaded all the final point-by-point reply to the comments of both referees, the marked-up manuscript version (changes made to text are written in red), and the revised files. In addition, we have included following major changes:

(1) Following the suggestion by Referee 1, a new table has been added (now Table 2). This table present yearly fluxes of diatoms.
(2) As part of the statistics performed, we added a new Figure (now Fig. 5=correlogram).
(3) Following Referee 2's and your own comments late Oct, we discuss the possible impact of the North Atlantic Oscillation (NOA). This is now addressed in l. 409-426.

We greatly appreciate the helpful reviewers' insights and your comments last October. We hope that this revised version will merit your positive consideration and the editorial requirements of *Biogeosciences*.

Best regards,

Oscar E. Romero
on behalf of co-authors

**A two-decades (1988-2009) record of diatom fluxes in the Mauritanian coastal upwelling: Impact of low-frequency forcing and a two-step shift in the species composition (bg-2020-336)**

Authors = Oscar E. Romero, Simon Ramondenc and Gerhard Fischer

Final response to Referee 1's comments
As required by BG, the response to the Referees is structured in the following sequence: (1) comments from Referee 1 (RC1) and (2) *authors' comments (AC)*.

**Comments from Referee #1**

RC1: Line 25: Please include AMO between brackets the first time it is mentioned in the text.
*AC: This has been corrected  (l. 25).*

RC1: Line 91: I believe authors could go a little bit further and state that this is the longest diatom time series sediment trap record of the world's ocean.
*AC: the sentence has been rephrased (l. 88-89).*

RC1: Material and methods Line 105 Since there are several gaps in the sediment trap record, authors could provide the number of days sampled during the 19-year record  (i.e. the proportion of days sampled versus the total number of days). This would help the reader to have a better idea of the gaps in the record.
*AC: this is now mentioned in l. 104-106.*

RC1: Line 108: Authors should be more specific and specify the depth range of the position of the sediment traps during their study in the text (i.e., not only in Table 1).
*AC: the depth range of the traps is now included in the revised version (l. 109-110).*

RC1: Line 111. While I agree with this statement, there were two mooring deployments with sediment traps deployed at 700 m and therefore their collection efficiency could have been compromised. Since the collection interval of one of these deployments coincided with an strong ENSO event, it is important that authors discuss in the text the possibility of collection efficiency issues during these intervals.
*AC: This issue is now addressed in l. 465-474.*

RC1: Line 156. Could authors provide annual diatom valve estimates for the years with the most complete records? Even a rough estimate of the annual fluxes at this site would be useful for the specialized reader in order to be able to compare the diatom fluxes of this site with other regions of the global ocean.
*AC: this helpful suggestion of R1 is now addressed in l. 187-290. We present the new Table 2 which contains yearly fluxes of total diatoms for 13 calendar years. We believe that a table showing the yearly fluxes will be more useful for other scientists than an additional figure.*

RC1: Results - Could authors provide a rough estimation, i.e. average daily and/or annual fluxes, for radiolarian and silicoflagellates fluxes? This would help the reader to understand the contribution of both groups in relation to diatoms. Also, as mentioned before, I would suggest to provide annual estimates in order to facilitate the comparison of the diatom valve fluxes of this site with other regions of the global ocean.
*AC: although we agree that fluxes of silicoflagellates and radiolarians can be of interest for scientists working on marine siliceous plankton, we emphasize that the focus of our long-term trap record is on the diatom fluxes and the species-specific composition of the assemblage. There is also a methodological aspect to*

*consider: the use of permanent slides for diatom and silicoflagellates census does not allow the proper quantification of radiolarian skeletons. Due to the low volume used in the preparation of the permanent slides for diatom counts, the low absolute concentration of radiolarian skeletons is too low for reliable radiolarian census.*

RC1: Please avoid the use of acronym TDF, i.e. write the name in full.
*AC: The acronym is now avoided and the full name (=total diatom flux) is used throughout the MS.*

RC1: 266 "the highest"
*AC: The sentence has been rephrased and reads now: "Fluxes in spring and summer show the highest amount of above-the-average total diatom concentration." (l. 282-283).*

RC1: 367 "concluded"
*AC: the sentence has been corrected to: "Using observational data and model experiments, Wang and Zhang (2013) concluded…" (l. 394-395).*

RC1: Line 385 Please specify/repeat when this change occurs here.
*AC: the corrected sentence reads now: "An extraordinary feature of the multiyear dynamics of diatom populations at the CBmeso site is the sharp shift in the species contribution in May and 2002 (Fig. 2b)." (l. 424-425).*

RC1: Line 429 "5.1.2 The occurrence of the strong 1997 ENSO and the response of the diatom community off Mauritania" The intense ENSO event registered by the traps coincides with the use of sediment trap record collected at substantially shallower depth than most of the other deployments. According to Table 1 the sediment trap from deployment "CBmeso8 upper" was placed at around 700 m while most of the traps used in the experiment were placed at > 3000 m (with some exceptions). The collection area of the shallower sediment trap and collection efficiency of the "CBmeso8 upper" could be different than the other records, and therefore it could have affected the composition of the diatom assemblage collected during this interval. Authors should discuss this point in the text.
*AC:  this issue is now addressed in l. 465-474. We believe that the strong resemblance in the species-specific composition of the diatom assemblage of both traps (highest contribution of diatoms associated with waters of moderate to low nutrient content), without any significant percentage shift, delivers sound evidence on the reliability of the diatom data at CB8 and CB9. Fischer et al. (2019, Global Biogeochemical Cycles, 33, 1100, https://doi.org/10.1029/2019GB006194) previously argued that the collection area of the lower CBmeso trap is larger that the one of the upper.  Only when the chlorophyll filament extends further out of the Mauritanian coast, particles reach also the upper traps.*

RC1: Line 455 Authors could also cite the possible impact of strong ENSO events on the Mediterranean diatom fluxes as reported by Bárcena et al. (2004) and Rigual-Hernández et al. (2013).
*AC:  Both articles are discussed in the revised version. (l. 506-507).*

RC1: Figure 4. The graphs in this figure are too small for proper visualization. Please increase the size of the graphs.
*AC: the revised version includes a larger and better resolved file of Figure 4.*

**A two-decades (1988-2009) record of diatom fluxes in the Mauritanian coastal upwelling: Impact of low-frequency forcing and a two-step shift in the species composition** (bg-2020-336)

Authors = Oscar E. Romero, Simon Ramondenc and Gerhard Fischer

Final response to Referee 2's comments

As required by BG, the response to the Referees is structured in the following sequence: (1) comments from Referee 2 (RC2) and (2) *authors' comments (AC)*.

**Comments from Referee #2**

RC2: (Frankcombe et al., 2010) with the presented dataset the authors can only check on how major shifts between positive and negative states of AMO, occurred within the period of this record, affect the Canary Upwelling system, but not its fully and longterm effect on the system. However, the NAO index of atmospheric circulation over Europe has a periodicity in the order of 7-8 years (Knut Lehre Seip et al., 2019) and the work of (Yamamoto and Palter, 2016) shows a clear relation between the NAO and the AMO, with northerly winds associated to a positive state of AMO and zonal winds to a negative state of AMO. As such, it would be interesting to verify the relation of your data with NAO variability, since upwelling is indeed a response to an atmospheric process. It would also have been nice to have a comparison with the upwelling index or northerly wind strength. Maybe through another statistical approach, something like cross-correlation?

*AC: We now provide an additional analytical test that supports our previous interpretations (correlogram, now Figure 5). We performed a correlation analysis with samples' score resulting from CA (Dim.1, Dim.2 and Dim. 3, which discriminates the diatom communities), climatic indexes (ENSO, NAO, AMO), diversity index (Shannon diversity) and fluxes (total diatom flux, freshwater diatom flux, Opal flux). As suggested by Reviewer 2, the correlogram shows a significant negative relationship between AMO and NAO. However, the goodness of fit between these climatic indexes was rather low ($R^2$ around 0.2). The correlogram also shows that the samples' score of first CA axis (Dim. 1, which discriminates the benthic from the other diatom groups) seems also impacted by the NAO, although with an exceptionally low $R^2$. However, the statistical tests (clustering, boxplot and the Kruskall Wallis approach) performed in the first submission do not show any relationship between diatom groups and the NAO. Conversely to the correlogram, our statistical approach analyses each community independently and does not compare one group with the others. Although both statistical approaches are correct, we believe that the correlogram method could induce some misunderstanding, leading to a certain degree of overestimation of NAO impact. We therefore conclude that AMO have a stronger impact on diatom communities off Mauritania than NAO has. However, we comment on the possible impact of NAO on the diatom community at site CBmeso and discuss the publications suggested.*

RC2: On which respects the effect of warming climate on the upwelling system and its primary production, you depart from the different conclusions reached by different studies, as presented in your introduction, to the proposal that your data is a different way of approaching the question. However, you conclude that your diatom data might be instrumental in distinguishing between climate-forced and intrinsic variability of the population of primary producers.

I have trouble with this statement, intrinsic variability is related to the basic needs of the organisms, so they will most probably change in function of the changes imposed on the system both by global and regional processes that in the end will also react to climate forcing!

*AC: this sentence has been rephrased as follows: Our 1988-2009 data set contributes to distinguish the impact of low-frequency climate forcings in the northeastern Atlantic and will be especially helpful for establishing the scientific basis for forecasting and modelling future states of the Canary EBUE and its decadal changes. (l. 36-39)*

RC2: Furthermore, although it is important to understand the process behind your stunning increase in benthic diatoms, your record does not allow you to verify what happens in terms of the plankton production and assemblage evolution during this 20yr. Or does it? Can you deduce the effect of the benthic flux that obscures the total record, and explore the 20yr variability of the planktonic diatom flux and assemblages 'composition that reach the trap?

*AC: It is true that the dramatic shift in the species-specific composition of the diatom assemblage in May 2002 does not imply any dramatic change in the absolute values of the total diatom concentration nor it translated into any significant changes of the biogenic silica (=opal) fluxes (see also Romero et al., 2017, Prog. Oceanogr. 159, 131). This observation also matches previous work at site CBmeso (Fischer et al., 2016, Biogeosciences 13, 3203).*

RC2: There is a general problem with the way references are listed in the text they do not follow an alphabetical order nor the year of publication.

*AC: The citation of articles and book chapters follow BG Instructions to Authors. We checked throughout and corrected when necessary.*

RC2: What is the reason to use the term pelagial rather than pelagic? Although used for lakes I have not seen any paper that defends/justifies its use for the ocean environment.

*AC: this has been rephrased throughout the revised MS.*

RC2: The use of satellite images (composites for the n_ of years considered for each specific time interval / diatom phase) for comparison would also be important to verify the variability on the surface water and upwelling conditions.

*AR: three pictures of chlorophyll* a *concentration have been added to Figure 1. They depict the average concentration of chlorophyll* a *for winters 1997, 2002 and 2008, gained with SeaWIFs (for 1997) and MODIS (https://oceancolor.gsfc.nasa.gov/cgi/l3, details will be provided in the revised version of the MS). The high interannual variability is clearly seen.*

RC2: Different depths of trap deployment at some time intervals (Table 1) may influence the diatom assemblage encountered as a result of a different catching area and the different contribution of particles transported by intermediate nepheloid layers. This needs to be acknowledged and discussed especially because one of the periods coincides with the ENSO period.

*AC: This issue -also raised by R1- is addressed in the replies to Referee 1's comments.*

RC2: Are you assuming that the intensification of the shelf and slope poleward current favors an increase in production of the benthic community and maintenance of the means of downslope transport, or the existence of a stronger poleward current gives rise to a stronger suspension of the benthic forms and their downslope transport in higher quantities? This needs clarification and discussion.

*AC: The occurrence of benthic diatoms in the hemipelagic CBmeso trap represents a lateral transport signal. It is well-known that the dynamic Mauritanian coastal waters serve as a jet for cross-shelf particle transfer and it produces sporadic particle clouds, which are advected hundred kilometers offshore within intermediate and bottom-near nepheloid layers (Nowald et al., 2015) toward the hemipelagic of the low-latitude NE Atlantic (Fischer and Karakaş, 2009). Subsurface waters (200 to 300 m depth) might be the place of mixing processes of older, laterally-advected materials from the shelf by giant filament activity, with relatively fresh material derived from the open ocean surface (Fischer et al., 2009). In addition to the nepheloid layer-mediated transport, the dynamics of water masses related to the existence of the canyon system off Mauritania might have contributed to the enhancement of transport from shallow water upon the trap site CBmeso. We speculate that the dramatic increase of benthic diatoms in the hemipelagic environment might be due to the intensification of the shelf and slope poleward transport upon deeper waters. Cross shelf particle transfer is not only restricted to the CC-EBUEs but is a general feature of these ecosystems (e.g. in the Californian EBUE, e.g. Barth et al. 2002, JGR 107)*

RC2: Pg. 3, Ln. 78 – The authors suggest that a different approach for the characterization of multiyear to interdecadal upwelling intensity in EBUEs is by assessing fluxes of particulates and microorganisms as captured by continuous sediment trap experiments."

*AC:  a vast majority of the previous studies on the long-term variability of productivity and upwelling intensity along the north-western African margin follows different approaches than the one of our study. Approaches previously used for the characterization of interannual upwelling variations mainly are velocity and directions of winds, annual wind stress, and Ekman transport. By stating that "A different approach for the characterization of multiyear to interdecadal trends in EBUEs is assessing fluxes of particulates and microorganisms as captured by continuous sediment trap experiments" (l. 77-79), we emphasize the fact that observational data based on interannual trap experiments are rare and represent a different approach to the study of possible links between variability of the microorganisms community, upwelling variations and the impact of low-impact climate and oceanographic forcing.*

RC2: Although you can assume that the flux of planktonic organism blooming in surface waters as a result of upwelling intensity, we are also aware that the nutrient content of the upwelling water is determinant for the size of the blooms as well as for the type of phytoplankton community. As such, bloom size and consequently microorganism fluxes could also reflect shifts in the upwelling source water associated with latitudinal shifts for example, rather than variations in upwelling intensity.
In fact, in this study besides the physical setting it is important to also consider the chemical (nutrient) and biological setting.

*AC: it is true that the occurrence of diatom populations (or those of any other organisms) at the CBmeso site is the result of the interaction of several processes acting in different timescales. The fact that the shift in the species-specific composition of the diatom assemblage in May 2002 is not paralleled by either an increase or decrease of total diatom and/or biogenic silica flux suggests that the intensity of upwelling per se did not significantly changed, nor an increase in DSi availability occurred after May 2002 in waters overlying site CBmeso.*

RC2:  Pg. 6, Ln. 103 – The SACW occurs in layers between 100 and 400 m depth at the Banc d'Arguin and off Mauritania.
*AC: the sentence has been corrected (l. 205-206).*

RC2: Pg. 8, Ln 250-252 – ENSO appears to be modulated by AMO, check Levin et al, (2017) or Chen et al., 2019 or Zhang et al., (2019).
*AC: Levine et al. (2017) and Zhang et al., (2019) are discussed in the revised version.*

RC2: Pg. 9. Ln. 301 – 302 – The list of species presented do correspond to marine plankton forms that although not thriving in the highly productive and colder coastal upwelling systems, and more likely to be found in warmer waters, they are also not characteristic or real oligotrophic waters.
*AC: in addition to other peer-reviewed publications, we base the grouping of diatom species found in the CBmeso trap samples on our almost 20-year continuous research of the temporal dynamics of diatom populations and their biogeographical occurrence. Throughout the years, we have learnt that the species listed as 'open-ocean taxa' are typical of ocean waters of low content of dissolved silica (DSi). From this point of view, we are confident in characterizing the open-ocean diatoms (as listed in Table 3 of our MS) as typical of oligotrophic waters. Other studies along the western African margin have used the same species characterization as we do here (Nave et al., 2001; Abrantes et al., 2002; Crosta et al., 2012).*

RC2: Pg. 10, Ln. 323- 324 - The impact of the environmental variables on diatom communities was investigated by simple comparison using the samples clustering and the forcing values associated (Fig 4). You are not using the forcing values, but rather the value of an index that is considered to define the

coherent mode of natural variability occurring in the north Atlantic. Changes in this mode will have an impact on the circulation at your study site and be considered a forcing factor for your specific process.
*AC: We agree with Referee 2 in that we used climate indexes, which is a proxy of the direct environmental forcing. We did not use highly-resolved environmental data (e.g., DSi content) because they are not available for the complete time series.*

RC2: Pg. 10, Ln. 329 – Please specify tendency of the gradient.
*AC: it has been re-phrased and reads as follows: In addition, a gradient in the Shannon diversity index of the diatom populations (Fig. 4c) is observed with predominant low values (1.7-2.5) corresponding to benthic (=group 4), intermediate values (2.7-3) for coastal planktonic (=group 3) and high values (3.1-3.45) in samples dominated by coastal upwelling and open-ocean populations (=groups 2 and 1) (pairwise Wilcoxon rank sum test; p-value<0.05). (l. 352-356)*

RC2: Pg. 10, Ln. 335 – Mentioned figure should be included as a supplementary figure.
*AC:  Figure 3 highlights our statistical approach to define which diatom communities dominate our samples and the time series of their respective dominance instead of doing it visually. Since this figure is also causally related to Figure 4, we do believe that Figure 3 should be kept as part of the MS figures and does not need to be transferred to Supplement.*

RC2: Pg. 10, Ln 337 - the benthic diatom *D. surirella* decreased the diversity, although it also seems to be promoted determined by AMO strengthening. In the same way, the  second CA axis samples scores are positively correlated with TDF, which confirms that coastal upwelling diatoms seems to promote define the TDF.
*AC: it has been re-phrased and reads as follows: 'Given that the first CA is positively driven by the benthic group, this confirms the outstanding dominance of the benthic diatom* D. surirella *after May 2002, which also appears linked to the strengthening of AMO. In the same way, the second CA axis is positively correlated with total diatom flux also confirms that coastal upwelling diatoms deliver large numbers to the total diatom valves.' (l. 377-379)*

RC2: Pg.11, Ln 352 - Based on outstanding shifts in the species-specific composition of the diatom assemblage occurred throughout the study interval (Fig. 2b).
*AC: it has been re-phrased and reads now: 'Based on outstanding shifts in the species-specific composition of the diatom assemblage occurred throughout the study interval (Fig. 2b), we propose three main intervals in the multiyear evolution of populations and discuss them in view of mayor environmental forcings:…' (l. 377-379)*

RC2: Pg. , Ln. 360 - Based on the long-term trends of our data and their statistical analysis (Figs. 2-5), we suggest that the proposed intervals were the response of the diatom populations to the impact of low frequency forcing on the Canary upwelling system. To be correct, the upwelling system is the one that responds to the low frequency forcing. Diatom assemblages reflect hydrographic and nutrient availability brought up by the upwelled source waters.
*AC: It is true that the upwelling in the Canary EBUE responds to low-frequency climate impact. By studying the diatom populations, we did not, however, directly characterize long-term variability of upwelling intensity off Mauritania as studies quoted in the Introduction of our first submitted version did (l. 66 th 77). Therefore, we believe that the sentence as written is correct.*

RC2: Figure 4: Comparison of (a) clusters extracted from multivariate analysis with the environmental forcing variables (a1: Total diatom flux; a2: AMO; a3: Shannon diversity). Besides being too small and difficult to see, total diatom Flux and Diversity are not forcing variables. They all reflect the community adaptation to the regional conditions resulting from the forcing factor(s).

*AC: it has rephrased (l. 348-350). NAO, AMO, ENSO and the diversity index Shannon-Weaver are indices while the total diatom flux is a variable. This was wrongly described in the original submission. The file resolution of Fig. 4 will be enlarged.*

RC2: References

Chen, S., Song, L. & Chen, W.: Interdecadal Modulation of AMO on the Winter North Pacific Oscillation-Following Winter ENSO Relationship. Adv. Atmos. Sci. 36, 1393– 1403, 2019. https://doi.org/10.1007/s00376-019-9090-1

Frankcombe, L. M., Heydt, A. v. d., and Dijkstra, H. A.: North Atlantic Multidecadal Climate Variability: An Investigation of Dominant Time Scales and Processes, Journal of Climate, 23, 3616-3638, 2010.

Levine, A. F. Z., M. J. McPhaden, and D. M.W.Frierson: The impact of the AMO on multidecadal ENSO variability, Geophys. Res. Lett., 44, 3877–3886, 2017. doi:10.1002/ 2017GL072524.

Knut Lehre Seip, Øyvind Grøn and Hui Wang: The North Atlantic Oscillations: Cycle Times for the NAO, the AMO and the AMOC. Climate, 2019, 7, 43; doi:10.3390/cli7030043

Yamamoto, A. and Palter, J. B.: The absence of an Atlantic imprint on the multidecadal variability of wintertime European temperature, Nature Communications, 7, 10930, 2016.

Zhang, W., X. Mei, X. Geng, A. G. Turner, and F. Jin: A Nonstationary ENSO– NAO Relationship Due to AMO Modulation. J. Climate, 32, 33–43, 2019. https://doi.org/10.1175/JCLI-D-18-0365.1.

*AC: we are grateful for these references. Most of these publications are now discussed in the revised version (l. 244-254 and l. 409-423).*